# Amorphous nickel-cobalt complexes hybridized with 1T-phase molybdenum disulfide via hydrazine-induced phase transformation for water splitting

Haoyi Li[1], Shuangming Chen[2], Xiaofan Jia[3], Biao Xu[4], Haifeng Lin[1], Haozhou Yang[1], Li Song[2] & Xun Wang[1]

Highly active and robust eletcrocatalysts based on earth-abundant elements are desirable to generate hydrogen and oxygen as fuels from water sustainably to replace noble metal materials. Here we report an approach to synthesize porous hybrid nanostructures combining amorphous nickel-cobalt complexes with 1T phase molybdenum disulfide ($MoS_2$) via hydrazine-induced phase transformation for water splitting. The hybrid nanostructures exhibit overpotentials of 70 mV for hydrogen evolution and 235 mV for oxygen evolution at 10 mA cm$^{-2}$ with long-term stability, which have superior kinetics for hydrogen- and oxygen-evolution with Tafel slope values of 38.1 and 45.7 mV dec$^{-1}$. Moreover, we achieve 10 mA cm$^{-2}$ at a low voltage of 1.44 V for 48 h in basic media for overall water splitting. We propose that such performance is likely due to the complete transformation of $MoS_2$ to metallic 1T phase, high porosity and stabilization effect of nickel-cobalt complexes on 1T phase $MoS_2$.

[1] Key Lab of Organic Optoelectronics and Molecular Engineering, Department of Chemistry, Tsinghua University, Beijing 100084, China. [2] National Synchrotron Radiation Laboratory, CAS Center for Excellence in Nanoscience, University of Science and Technology of China, Hefei 230029, China. [3] Department of Chemistry, University of Virginia, Charlottesville, Virginia 22904, USA. [4] Department of Chemical and Biological Engineering, Iowa State University, Ames, Iowa 50010, USA. Correspondence and requests for materials should be addressed to X.W. (email: wangxun@mail.tsinghua.edu.cn).

Hydrogen ($H_2$) has attracted extensive attention for decades as an environmentally friendly energy source[1–4]. Electrochemical or photo-electrochemical water splitting is a convenient method to generate hydrogen, converting electricity to chemical fuels for energy storage and transport[5,6]. However, the dynamically unfavourable nature of water splitting is regarded as the bottleneck, in which both the hydrogen- and oxygen-evolution reactions (HER and OER) need high overpotentials for activation. Currently, Pt alloys and Ir/Ru oxides are regarded as the state-of-the-art electrocatalysts for HER and OER respectively, but cost and scarcity are the barriers for the scale-up utilization of these noble-metal catalysts in industrial deployment[7–9]. The development of catalysts for HER and OER with non-noble materials has achieved great success. Molybdenum disulfide ($MoS_2$) is one of the most promising candidates for HER electrocatalysts representing transition metal dichalcogenides, which has the possibility to replace Pt-based electrocatalysts for practical applications[10–15]. For OER, low overpotentials and moderate durability have been exhibited by transition-metal (especially nickel and cobalt) sulfides[16,17], selenides[18,19], oxides[20,21], phosphides[22] and layered double hydroxides[23]. However, it is quite difficult to obtain high concentrations of $H^+$ and $OH^-$ simultaneously to motivate HER and OER because they follow the rule of $[H^+] \cdot [OH^-] = 10^{-14}$. Meanwhile, if both catalysts were employed for electrolysis, the cost would increase because of the complicated process of manufacturing electrodes. Thus, it is quite challenging to develop bifunctional electrocatalysts for HER and OER in one electrolyte. Although some progress has been made in this field[24–29], more efforts should be devoted to designing the catalysts and enhancing their performance to control the industrial cost and lower the energy consumption. A recent work illustrated that $MoS_2/Ni_3S_2$ heterostructures designed by interface engineering show excellent performance for overall water splitting, which could synergistically chemisorb hydrogen and oxygen-containing intermediates[29]. This material provides the possibility for fabricating new and efficient electrocatalysts by hybridizing nickel-cobalt-based (Ni-Co-based) compounds with $MoS_2$.

Herein, we present a facile strategy to synthesize porous hybrid nanostructures combining amorphous Ni-Co complexes with 1T phase $MoS_2$ (denoted as PHNCMs) through hydrazine-inducing, which have highly active and ultra-stable electrocatalytic performances towards HER and OER. Notably, the PHNCMs achieve overpotentials of 70 mV for HER and 235 mV for OER at 10 mA cm$^{-2}$ and fast kinetics shown by low Tafel slope values of 38.1 and 45.7 mV dec$^{-1}$ for HER and OER respectively. Meanwhile, this material holds an overvoltage of 1.44 V to reach a current density of 10 mA cm$^{-2}$ for 48 h operation without degradation for overall water splitting.

In this work, we introduce hydrazine hydrate (HZH) into the reaction system to regulate the crystallization of Ni-Co-based compounds and the phase of $MoS_2$ (Fig. 1, see Methods for synthetic details). With the increasing amount of HZH, Ni-Co-based compounds are changed to amorphous complexes from a partial metallic state and $MoS_2$ is completely converted to metallic 1T phase. We propose that the phase transformation of $MoS_2$ is attributed to the enrichment of amorphous Ni-Co complexes with electron-donor ability of hydrazine because of the stabilization effects of the complexes on 1T phase $MoS_2$. Metallic 1T phase $MoS_2$ can facilitate the electrode kinetics, increase the electric conductivity of the electrocatalysts and proliferate the catalytic active sites[30,31]. Concurrently, porous nanostructures can create more catalytic active sites and improve the mass transport and gas permeability effectively in the process of water splitting[32,33]. Moreover, hydrazine involved Ni-Co complexes have the amine residues in the second coordination sphere where intramolecular proton transfer takes place preferentially, which is beneficial to lower the overpotential of electrocatalytic $H_2$ evolution[34,35]. The large quantity of Ni-Co complexes in the PHNCMs is helpful to promote the catalytic activities of HER and OER simultaneously. Therefore, we fabricate the PHNCMs to integrate the advantages of every component in electrocatalysis, resulting in the enhancement of performance for overall water splitting.

## Results

**Synthesis and characterization of PHNCMs.** We first synthesized the Ni-Co hydroxides ultrathin nanosheets (NCUNs) to provide the precursors and templates for the following synthesis of PHNCMs. The pristine NCUNs showed ~4 nm uniform thickness and 30–150 nm diameters with circular shape according to atomic force microscopy (AFM) and transmission electron microscopy (TEM) images (Fig. 1a–c). The crystalline structure of NCUNs was demonstrated through X-ray diffraction pattern (Supplementary Fig. 1a). The X-ray photoelectron spectroscopy (XPS) spectrum of O 1s orbital in NCUNs (Supplementary Fig. 1b) further confirmed the hydroxide feature of NCUNs owing to the peak position at 531.5 eV (ref. 36).

After NCUNs reacted with ammonium tetrathiomolybdate (($NH_4)_2MoS_4$) in N,N-dimethylformamide (DMF) solvent with different amount of HZH, the PHNCMs were formed, which exhibited irregular nanosheet-like structures with uneven surface and increasing porosity by TEM and high-resolution TEM (HRTEM) images (Fig. 2d–g and Supplementary Fig. 2, PHNCMs with no HZH, 0.05 ml of HZH, 1 ml of HZH and 2.5 ml of HZH denoted as 0H-PHNCMs, 0.05H-PHNCMs, 1H-PHNCMs and 2.5H-PHNCMs). The hybrid nanostructures maintained the nanosheet morphology, as compared to the products synthesized from corresponding metal acetates directly in the same reaction system as 0H-PHNCMs (Supplementary Fig. 3). It suggested that

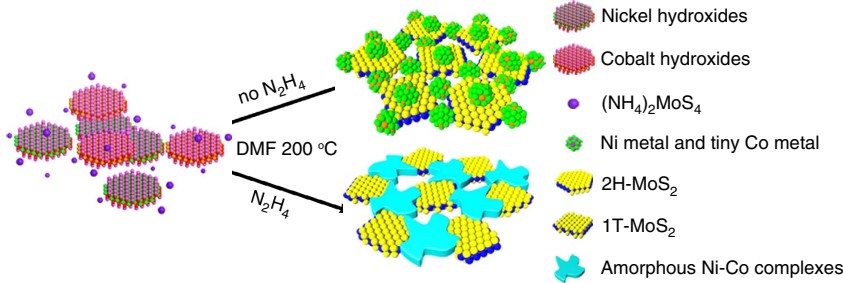

**Figure 1 | Schematic representation of the formation of PHNCMs.** Upper route illustrates the synthesis of Ni metal and tiny Co metal hybridized with $MoS_2$ in the blended phase of 2H and 1T without HZH using NCUNs and ($NH_4)_2MoS_4$ as the precursors by a solvothermal method. Lower route demonstrates the synthesis of the hybrid nanostructures of amorphous Ni-Co complexes and 1T phase $MoS_2$ with large amount of HZH using the same precursors as upper route.

NCUNs were excellent templates for building hybrid nanosheet-like structures. HZH could decompose into nitrogen ($N_2$) and ammonia ($NH_3$) at high temperatures, after which $N_2$ and $NH_3$ bubbles acted as the gas templates resulting in the formation of pores. We inferred that the number and size of pores positively correlated to the quantity of HZH based on the TEM images (Fig. 2d–g) and the corresponding pore size distribution curves (Fig. 2h–k). When the quantity of HZH reached 2.5 ml, the number of pores increased markedly and pores with larger diameters (~10 nm) appeared. Meanwhile, specific surface areas

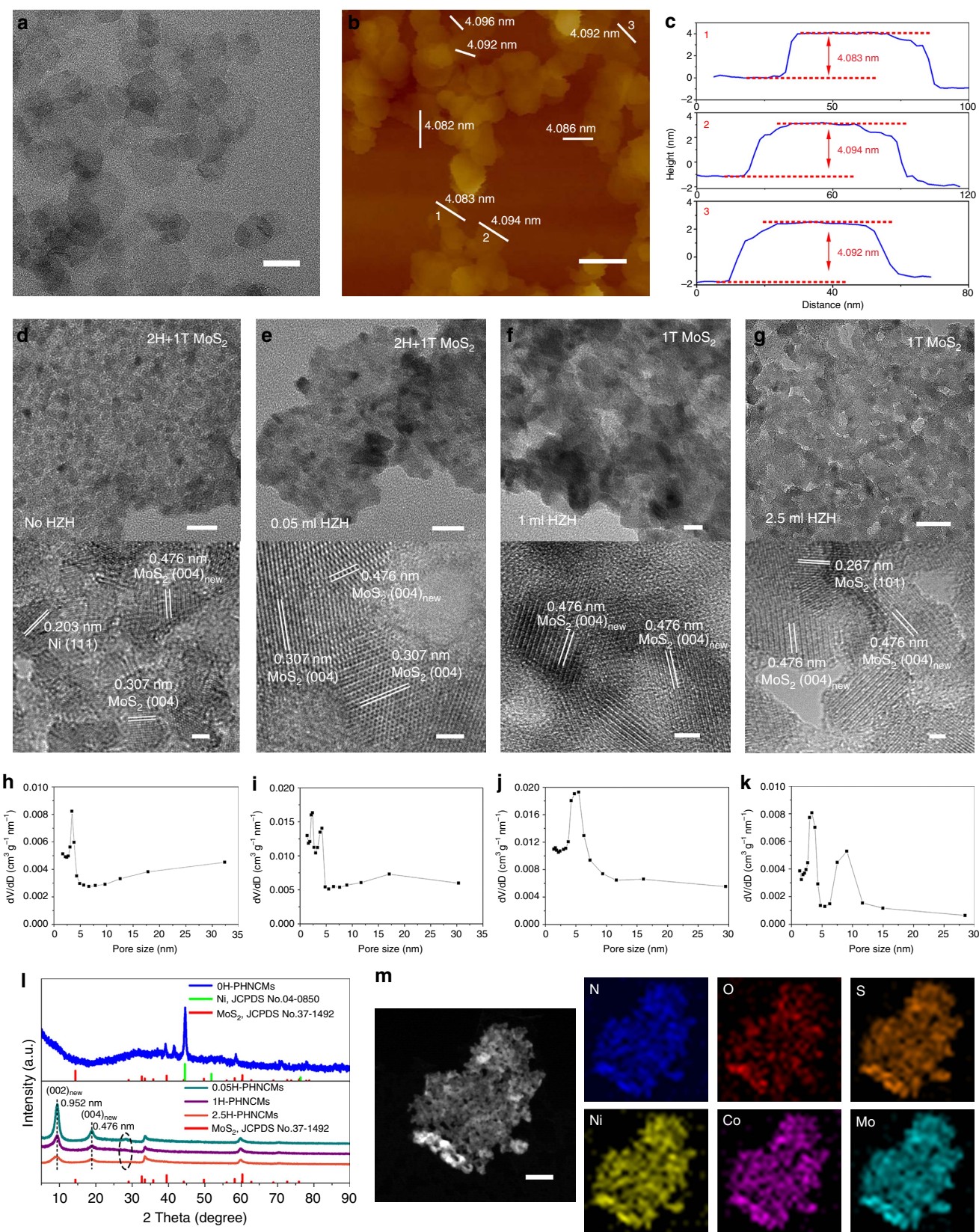

were obtained by the $N_2$ adsorption–desorption isotherms (Supplementary Fig. 4). 2.5H-PHNCMs had the largest specific surface area of $90.68 \, m^2 \, g^{-1}$ among the PHNCMs, leading to more active catalytic sites to enhance the performance of electrocatalysis. Furthermore, when the amount of HZH increased continuously to 3 and 5 ml (denoted as 3H-PHNCMs and 5H-PHNCMs), no big differences were observed in morphoogies, specific surface area and pore size distribution curves (Supplementary Fig. 5) as compared with that of 2.5H-PHNCMs.

Integrating the results of HRTEM images (lower images in Fig. 2d–g and Supplementary Fig. 6) and the corresponding X-ray diffraction patterns (Fig. 2l and Supplementary Fig. 7), Ni was changed to amorphous state from metallic state, new (002) and (004) planes of $MoS_2$ with the lattice distance of 0.952 and 0.476 nm were observed and pristine (004) planes of $MoS_2$ with the lattice distance of 0.307 nm disappeared gradually when the amount of HZH increased. The enlargement of interlayer spacing of $MoS_2$ indirectly suggested the existence of metallic 1T phase of $MoS_2$ in PHNCMs[37]. We further confirmed the state of Ni and Co in 0H-PHNCMs. The X-ray diffraction pattern (Supplementary Fig. 8) of Ni-only hybrid nanostructures synthesized in the same way as 0H-PHNCMs could match with the standard patterns of $MoS_2$ and Ni metal and that of Co-only hybrid nanostructures was only in accordance with the standard pattern of $MoS_2$. It is well known that HZH is a strong reductant as well as a strong coordination ligand[38–40]. Therefore, the introduction of HZH to the reaction system led to the preferential formation of hydrazine coordinated Ni-Co complexes. According to the X-ray diffraction matching consequences, the crystalline structure in PHNCMs with HZH just agreed with $MoS_2$, which indicated that Ni and Co existed in the amorphous states. The scanning TEM (STEM) images and energy-dispersive X-ray (EDX) elemental mapping spectra of PHNCMs (Fig. 2m and Supplementary Fig. 9) exhibited the elemental composition and uniform distribution of the six elements in PHNCMs. The quantity of nitrogen increased significantly in PHNCMs with HZH as compared with that in 0H-PHNCMs that was proven by elemental proportion analyses of nitrogen (Table 1). The increasing proportions of nitrogen revealed that it was likely for Ni and Co to form hydrazine coordinated amorphous complexes. Therefore, it was quite possible that PHNCMs with large amount of HZH were composed of amorphous Ni-Co complexes and 1T phase $MoS_2$, which was further revealed in the following characterization.

**Analyses for atomic structure of the PHNCMs.** In-depth analyses to determine the atomic structure of PHNCMs were made by several characterization methods. First, we investigated the bonding situation of Ni and Co in PHNCMs through the synchrotron-radiation-based extended X-ray absorption fine structure (EXAFS). Structural parameters of the bond lengths, the coordination numbers, the Debye-Waller factors and the comparison between experimental data and fitting curves for Ni, Co and Mo K-edge were summarized in Supplementary Table 1 and 2 and Supplementary Figs 10–12. The normalized X-ray absorption near edge structure (XANES) spectra of Ni and Co K-edge of the PHNCMs (Supplementary Fig. 13a and 13b) exhibited the content sequences of Ni and Co metal. According to the intensity of white line and pre-edge peaks, as compared with the standard samples, they were 0H-PHNCMs $\gg$ 1H-PHNCMs > 0.05H-PHNCMs > 2.5H-PHNCMs and 0H-PHNCMs > 1H-PHNCMs > 2.5H-PHNCMs > 0.05H-PHNCMs, respectively for Ni and Co. Fig. 3a,b presented $k^3$-weighted Fourier transform (FT) profiles of XANES at Ni and Co K-edge of PHNCMs and standard Ni foil, NiO, Co foil, CoO as contrastive samples. For 0H-PHNCMs, it was clearly seen that Ni-Ni bond with 2.48 Å length were the main existing forms in the real space. The peaks at ∼1.72 Å for Ni in the first shell of the three PHNCMs with HZH were attributed to the major contribution of Ni-O or Ni-N bonds. Meanwhile, the peaks for Co in the first shell of all the PHNCMs upshifted slightly as compared to the first peak position of standard CoO, which revealed that Co–O or Co-N bonds predominated but there was still a small quantity of Co-Co bond. Surface features of Ni and Co were probed by XPS measurement (Supplementary Fig. 14). For 0H-PHNCMs, the peaks at ∼853.09 and ∼870.3 eV stood for Ni metal in Ni-Co alloy and the small peak at ∼856.3 eV indicated that a little bit of Ni in oxidation state located on the surface, while the peaks at ∼778.5 and ∼780.7 eV were indexed to Co metal and Co in oxidation state[41,42]. Along with the increasing amount of HZH, peaks for both of metallic Ni and Co were attenuated and peaks of oxidation state at higher binding energies became stronger gradually. As Supplementary Fig. 15 shown, Co (III) and Ni (III) predominated and Co (II) and Ni (II) also existed on the surface of 2.5H-PHNCMs. However, there were still a small quantity of Co and Ni metal, the small peaks of which could be observed in the XPS spectra. This result corresponded well to the FT profiles of XANES at Ni and Co K-edge. During the synthesis of 2.5H-PHNCMs, hydrazine was added dropwise slowly with stirring at the room temperature before heating. In this case, Co and Ni ions preferentially coordinated with hydrazine to form complexes rather than being reduced. At the same time, a small amount of potassium hydroxide (KOH) was fed into the system to make $NH_3$ form and escape easily at a high temperature. The $NH_3$ would be a gas template to make sure the formation of porous nanostructures. So the quantity of KOH was not enough to cause the preformed Ni-Co-hydrazine complexes to be reduced to metallic states[39,40]. When Co and Ni coordinated with hydrazine to form complexes, it is easy for Co (II) and Ni (II) complexes to be oxidized to Co (III) and Ni (III) ones. Because the redox potential of hydrazine-coordinated complexes, $\psi^\theta(Co^{3+}/Co^{2+})$ and $\psi^\theta(Ni^{3+}/Ni^{2+})$,

**Figure 2 | Characterizations of the NCUNs and PHNCMs.** (**a**) TEM image (**b**) AFM image and (**c**) the corresponding line-scan profiles of NCUNs, showing NCUNs with ∼4 nm thickness and 30–150 nm diameters. (**d,e**) TEM (upper) and HRTEM (lower) images of 0H-PHNCMs and 0.05H-PHNCMs showing porous nanosheet-like structures and uneven surface. The lattice distance of 0.476 nm is indexed to new (004) lattice plane of $MoS_2$ as compared to pristine (004) planes with 0.307 nm of distance. The enlargement of lattice distance suggests that 1T phase $MoS_2$ exists in the 0H-PHNCMs and 0.05H-PHNCMs. With the increasing amount of HZH to (**f**) 1 ml and (**g**) 2.5 ml, TEM and HRTEM images exhibit the disappearance of pristine (004) planes, suggesting the complete transformation of $MoS_2$ to metallic 1T phase. The distances of 0.203 and 0.267 nm are consistent with the standard spacing of (111) planes in Ni metal and (101) planes in $MoS_2$, respectively. (**h–k**) Pore size distribution curves of 0H-, 0.05H-, 1H- and 2.5H-PHNCMs in accordance with TEM images in **d–g**. (**l**) X-ray diffraction patterns of PHNCMs. Lower patterns display the shifts of (002) and (004) peaks (corresponding to the lattice distances of 0.952 and 0.476 nm) and gradual disappearance of pristine (004) peaks at ∼29° of 2 Theta in the short dash circle, illustrating the increasing interlayer space of $MoS_2$. (**m**) STEM and EDX elemental mapping spectra of 2.5H-PHNCMs showing the porous nanostructures and uniform distribution of N (blue), O (red), S (orange), Ni (yellow), Co (pink) and Mo (cyan). Scale bars, 100 nm (**a,b**); upper, 20 nm (**d–g**, upper); 2 nm (lower); 50 nm (**m**).

| Table 1 | Percentage proportions of nitrogen in different PHNCMs. | | | |
|---|---|---|---|---|
| Samples | 0H-PHNCMs | 0.05H-PHNCMs | 1H-PHNCMs | 2.5H-PHNCMs |
| Ratios | 3.37% | 10.65% | 13.92% | 14.88% |

was decreased dramatically to the $10^{-1}$ from $10^0$ of magnitude, so that hydrazine-coordinated Co (III) and Ni (III) complexes could exist steadily. When the amount of HZH increased to 3 and 5 ml, Ni and Co in the oxidation states were observed quite similarly (Supplementary Fig. 14). The above analyses further indicated the existence of hydrazine-coordinated Ni-Co complexes in the PHNCMs with HZH.

Subsequently, we investigated the local bond length of Mo in PHNCMs through XANES (Supplementary Fig. 13c) to demonstrate the phase transformation of $MoS_2$. According to the FT profiles and bond lengths at Mo K-edge (Fig. 3c and Supplementary Table 2), the nearest Mo-Mo bonds of PHNCMs showed distinct decreasing lengths from 3.16 to 2.78 Å, which was in agreement with structural transformation from hexagonal to tetragonal phase[37]. For 0H- and 0.05H-PHNCMs, the peaks at 2.82 Å indicated that there was still some 2H phase $MoS_2$ in the hybrid nanostructures. In contrast, $MoS_2$ in 1H-PHNCMs and 2.5H-PHNCMs was completely transferred to 1T phase as the evident disappearance of Mo-Mo peaks at 2.82 Å for 2H phase.

XPS spectrum of Mo (Fig. 3d and Supplementary Fig. 16a) displayed the obvious shifts of peaks to lower binding energies with the increasing amount of HZH. For 1H-, 2.5H-, 3H- and 5H-PHNCMs, two characteristic peaks of Mo $3d_{5/2}$ and Mo $3d_{3/2}$ orbitals were located at ~228 and ~231.0 eV, which were much lower than that of the 2H phase counterparts (~229 and 232 eV)[30,43]. Similar downshifts of binding energies for S $2p$ orbitals were exhibited in Fig. 3e and Supplementary Fig. 16b. These downward movements of Mo $3d$ and S $2p$ peak positions further proved the formation of metallic 1T phase $MoS_2$. In addition, we attempted to demonstrate the reason behind phase transformation of $MoS_2$. We synthesized $MoS_2$ nanosheets with different amount of HZH in the same way as PHNCMs ($MoS_2$ with no HZH, 0.05 ml of HZH, 1 ml of HZH and 2.5 ml of HZH denoted as 0H-$MoS_2$, 0.05H-$MoS_2$, 1H-$MoS_2$ and 2.5H-$MoS_2$). TEM images (Supplementary Fig. 17a–d) showed the distinct nanosheet morphology of $MoS_2$ and the tendency to assemble spherically. The two peaks below 20° in the X-ray diffraction pattern (Supplementary Fig. 17e) revealed that the enlarged interlayer spacing emerged as compared to pristine $MoS_2$, which was in good agreement with that of the PHNCMs. A binding energy comparison of Mo was illustrated in the XPS spectrum (Fig. 3f) of the above a series of $MoS_2$ nanosheets. Inconspicuous downshifts (~0.25 eV) were observed for peaks of Mo $3d_{5/2}$ and Mo $3d_{3/2}$ orbitals between 0H-$MoS_2$ and 2.5H-$MoS_2$, which were far less than that in PHNCMs (~1 eV). Analogously, the peaks of S $2p$ orbitals in XPS spectrum (Supplementary Fig. 18) showed the weak downshifts (~0.2 eV). The above results indicated that 2.5H-$MoS_2$ was not in pure 1T phase and Ni-Co complexes were essential for the completely phase transformation of $MoS_2$.

It has been reported that electron donor is quite necessary in the phase transformation of $MoS_2$ from 2H to 1T phase[44]. In our case, hydrazine could be regarded as the electron donor to induce the phase transformation. However, if NCUNs were not introduced to the reaction system of 2.5H-PHNCMs, pure 1T phase $MoS_2$ could not have been obtained. Compared to 2H phase $MoS_2$, 1T phase $MoS_2$ has higher ground-state energy[45]. It is extremely possible that amorphous Ni-Co complexes play the

role of stabilization of 1T phase $MoS_2$, which could be further demonstrated by the following results.

**Electrochemical evaluation of the PHNCMs.** On the basis of the successful synthesis and comprehensive characterization, we examined HER and OER electrocatalytic activities of the PHNCMs. The as-prepared PHNCMs mixing with pure carbon (Vulcan XC72) were evaluated using a typical three-electrode system in 1 M KOH media. A saturated calomel electrode (SCE) was used as the reference electrode. It was calibrated with reversible hydrogen electrode (RHE) and the potentials were reported versus RHE (Supplementary Fig. 19). $MoS_2$ nanosheets (with 0.05 ml of HZH), NCUNs, commercial Pt/C and $IrO_2$/C catalyst were also measured for comparison. 2.5H-PHNCMs displayed excellent electrocatalytic activity for HER, which approached that of commercial Pt/C catalyst. As a consequence of polarization (Fig. 4a), 2.5H-PHNCMs only needed an over-potential of 70 mV to reach a current density of 10 mA cm$^{-2}$ that was much better than that of 0H-PHNCMs (87 mV at 10 mA cm$^{-2}$), not to mention 148 and 128 mV for 0.05H-PHNCMs and 1H-PHNCMs (Supplementary Fig. 20a). After linear fitting for the Tafel plot of 2.5H-PHNCMs, the slope was calculated to be 38.1 mV dec$^{-1}$ (Supplementary Fig. 21a), which was lower than that of 0H-PHNCMs (40.3 mV dec$^{-1}$), $MoS_2$ nanosheets (46.4 mV dec$^{-1}$) and NCUNs (89.4 mV dec$^{-1}$). Such a low Tafel slope value illustrated the superior HER kinetics of 2.5H-PHNCMs, further confirming that they were indeed excellent HER electrocatalysts. Electrochemical impedance spectroscopy (EIS) Nyquist plots (Supplementary Fig. 21b) manifested that 2.5H-PHNCMs had smaller reaction resistance as compared to other contrasting catalysts, suggesting that charge transfer was facilitated in 2.5H-PHNCMs. To assess the stability of 2.5H-PHNCMs, Chronoamperometric response (Fig. 4c) was recorded at a constant potential of −0.13 V versus RHE for 24 h. During the initial 3 h, the performance degraded; however, the current density increased for the remaining time. More efficient HER active sites might exist in the interfaces between $MoS_2$ and Ni-Co complexes. The 3 h reduction allowed more electrolyte to access the interfaces, obtaining the increasing HER current density during the following time[46]. To measure electrochemical active surface area (ECSA), we first scanned cyclic voltammetry (CV) cycles in the range of no Faradaic processes (Supplementary Fig. 22), obtained double-layer capacitance ($C_{dl}$) (Fig. 4e) and then converted it to ECSA. We found that 2.5H-PHNCMs had the largest ECSA of 807.5 cm$^2$ as compared to 0H-PHNCMs (502.5 cm$^2$), $MoS_2$ nanosheets (255 cm$^2$) and NCUNs (45 cm$^2$). The number of catalytic active sites could be determined roughly by ECSA. Therefore, 2.5H-PHNCMs possessed the maximum active sites for HER among all of the contrasts. Supplementary Fig. 23a showed the comparison between the theoretical amount of $H_2$ calculated from a chronopotentiometric response and the evolved quantity of $H_2$ experimentally measured from a gas chromatography in the process of HER over 2.5H-PHNCMs for 120 min. The experimental values were observed to extremely approach to the theoretical values. This result indicated that 2.5H-PHNCMs provided a Faradaic efficiency of ≈100% for the HER. It is a convincing evidence of water splitting, which means

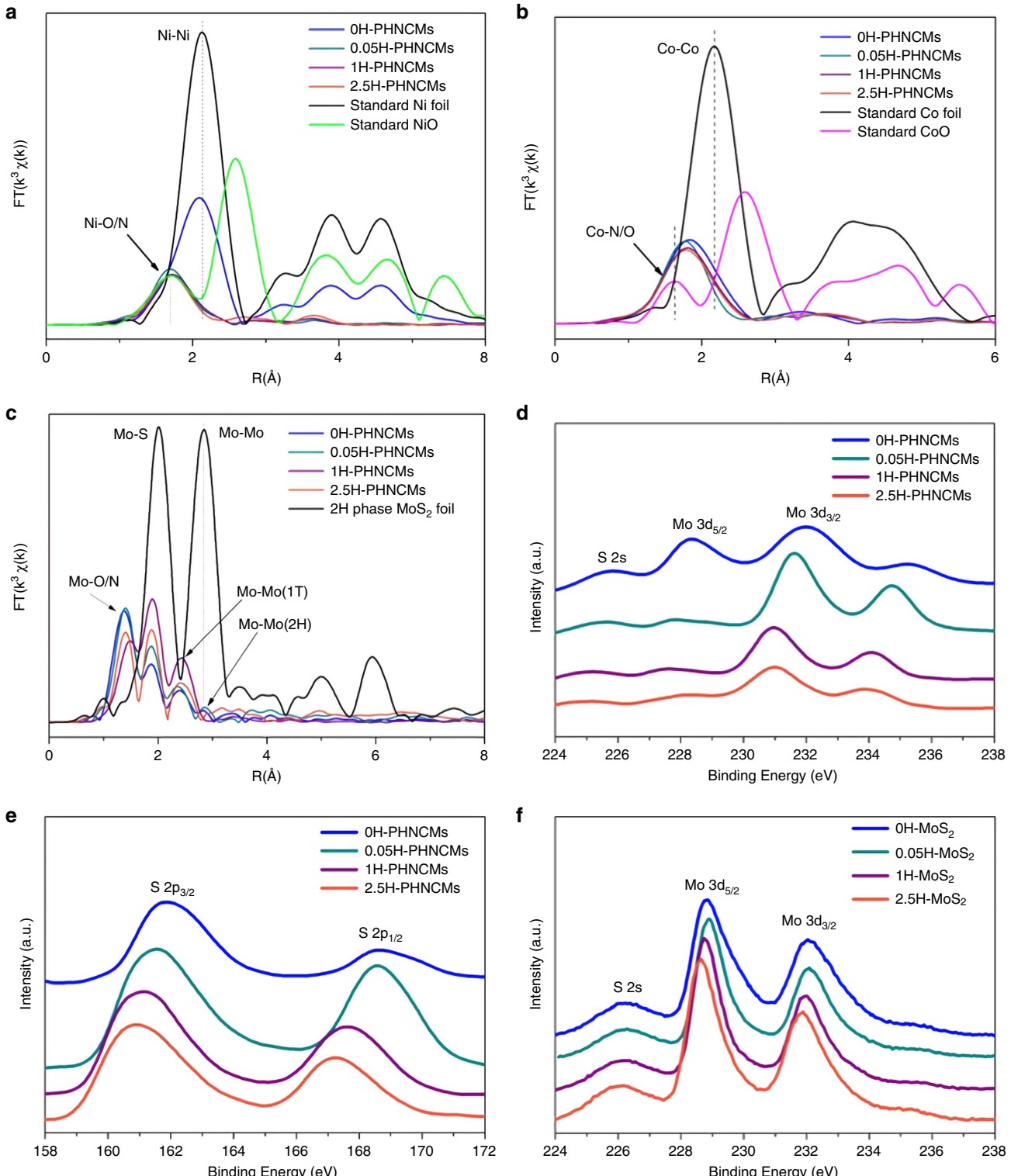

**Figure 3 | Characterization for atomic structure of the PHNCMs. (a–c)** The $k^3$-weighted FT spectra of XANES from EXAFS at the Ni, Co and Mo K-edge of the PHNCMs and Ni foil, NiO, Co foil, CoO, 2H phase $MoS_2$ foil as contrasting samples. A large quantity of Ni-Ni bond and tiny amount of Co-Co bond possess the major contribution in 0H-PHNCMs. Ni-O or Ni-N and Co-O or Co-N are the main bond for Ni and Co in PHNCMs with HZH. The third and fourth peaks at $\sim 2.40$ Å and $\sim 2.85$ Å of 0H-PHNCMs and 0.05H-PHNCMs for Mo-Mo bond indicate that 1T and 2H phase $MoS_2$ coexist in the samples. The disappearance of the fourth peaks at $\sim 2.85$ Å of 1H and 2.5H-PHNCMs illustrates the complete phase transformation of $MoS_2$ to metallic 1T phase. The smoothing XPS spectra showing the binding energies of **(d)** Mo and **(e)** S in the PHNCMs. Obvious downshifts ($\sim 1\,eV$) of Mo 3d and S 2p peak positions demonstrates the phase transformation from 2H to 1T phase. **(f)** The smoothing XPS spectrum of Mo 3d orbitals in $MoS_2$ synthesized in the same way as PHNCMs. With the increasing quantity of HZH, the peaks of Mo $3d_{3/2}$ and Mo $3d_{5/2}$ orbitals in $MoS_2$ have slight downshifts ($\sim 0.25\,eV$) to lower binding energies, suggesting that amorphous Ni-Co complexes were the main cause in the phase conversion of $MoS_2$.

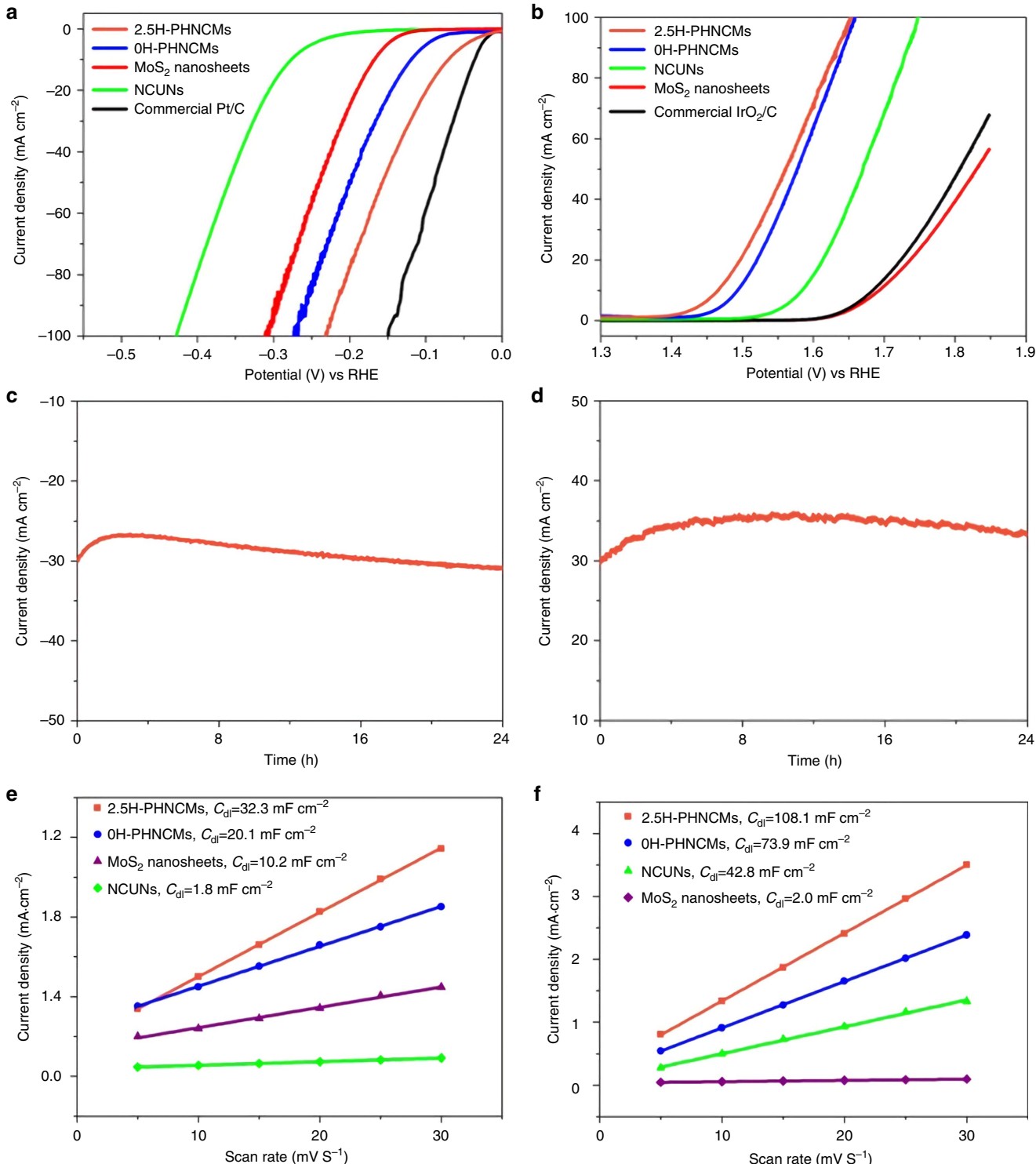

**Figure 4 | Electrocatalytic hydrogen and oxygen evolution of different catalysts.** (**a,b**) Polarization curves for HER and OER measured at a scan rate of 5 mV s$^{-1}$ in 1 M KOH electrolyte; (**c,d**) chronoamperometric responses (i-t) recorded on 2.5H-PHNCMs for 24 h at a constant applied potential of − 0.13 V versus RHE for HER and 1.53 V versus RHE for OER; (**e,f**) the fitting plots showing $C_{dl}$ for HER and OER.

that the cathodic currents over 2.5H-PHNCMs derived from the hydrogen evolution.

In addition to the excellent performance for HER electrocatalysis, 2.5H-PHNCMs were also efficient electrocatalysts for OER. During polarization process (Fig. 4b), 2.5H-PHNCMs could afford a current density of 10 mA cm$^{-2}$ with an overpotential of 235 mV and kept the activity for 24 h (Fig. 4d),

which outstandingly ranked as the best among reported OER catalysts[16–23] and was much better than that of commercial IrO$_2$/C catalyst. A performance comparison of PHNCMs for OER polarization (Supplementary Fig. 20b) was made and it was found that 2.5H-PHNCMs and 0H-PHNCMs showed better activities and the current densities of them tended to be the same after 1.6 V. At the same time, the Tafel slope value of

2.5H-PHNCMs (45.7 mV dec$^{-1}$) was the smallest among all of the tested catalysts (Supplementary Fig. 21c), implying that they had superior OER kinetics. Moreover, EIS Nyquist plots (Supplementary Fig. 21d) showed that they made electrons transfer easier according to the < 30 ohm reaction resistance. Similarly, we measured the $C_{dl}$ to estimate the ECSA of each sample for OER (Fig. 4d and Supplementary Fig. 24). Because of the larger proportion of Ni and Co as compared to Mo (Supplementary Table 3), PHNCMs had more catalytic active sites for OER than that for HER. 2.5H-PHNCMs possessed 2,702.5 cm$^2$ of ECSA, which was much larger than that of others (1,847.5 cm$^2$ of 0H-PHNCMs, 1,070 cm$^2$ of NCUNs and 50 cm$^2$ of MoS$_2$ nanosheets). For the Faradaic efficiencies for OER, the contrast between the measured amount of O$_2$ production and the theoretical values was shown in Supplementary Fig. 23b and Supplementary Table 4. It was found that the experimental values were a little less than the theoretical values due to the complex four-electron reaction process of OER, whose kinetics was quite unfavourable. The Faradaic efficiencies at the different periods were also calculated and the average Faradaic efficiency was 91.23%.

We also measured the electrocatalytic performances for HER and OER over 3H- and 5H-PHNCMs and made a comparison with 2.5H-PHNCMs. It was observed that the three PHNCMs exhibited quite similar activities for HER and OER electro-catalysis (Supplementary Fig. 25). Significant evaluation indexes were summarized in Supplementary Table 5.

Furthermore, We made a comparison of XPS spectra (Supplementary Fig. 26a,b) of Mo 3$d$ orbitals in 2.5H-PHNCMs and 2.5H-MoS$_2$ before and after 1,000 OER cycles. The positions of the two characteristic peaks of Mo 3$d_{5/2}$ and Mo 3$d_{3/2}$ orbitals in 2.5H-PHNCMs did not change after 1,000 OER cycles. However, upshifts (∼0.4 eV) were observed for the peaks of Mo 3$d_{5/2}$ and Mo 3$d_{3/2}$ orbitals in 2.5H-MoS$_2$ after 1,000 OER cycles, which revealed in situ electrochemical oxidation of 2.5H-MoS$_2$ during OER process. Besides, new (002) and (004) planes with the enlargement lattice distance of 0.952 and 0.476 nm were observed in X-ray diffraction pattern (Supplementary Fig. 26c) and HRTEM image (Supplementary Fig. 26d) of 2.5H-PHNCMs after 1,000 OER cycles, which could further confirm that MoS$_2$ in 2.5H-PHNCMs remained 1T phase after 1,000 OER cycles. The above illustration demonstrated that amorphous Ni-Co complexes played a role of stabilization of metallic 1T phase MoS$_2$.

Such excellent electrocatalytic performance should be attributed to the contribution of both components in 2.5H-PHNCMs. On one hand, metallic 1T phase MoS$_2$ can increase the electric conductivity and catalytic active sites[30,31], which was proven by the low electrochemical reaction resistances and large ECSAs. On the other hand, large proportions of Ni and Co result in marvelous electrocatalytic OER activity. Here we propose a possible mechanism of intramolecular proton transfer in hydra-zine coordinated Ni-Co complexes (Supplementary Fig. 27) that beneficially lowers the overpotential of HER[34,35]. Our Ni-Co complexes have amine residues in the second coordination sphere, which could be a hydrogen-exchanging site. As shown in Supplementary Fig. 15, Co (III) and Ni (III) predominated in 2.5H-PHNCMs. The catalytic trivalent metal linking to hydrazine obtains two electrons to become the monovalent metal. Then, two protons combine with the monovalent metal and nitrogen atom of the amine residue in the second coordination sphere to form a metal hydride and a –NH$_3^+$ group. Subsequently, the two protons tend to combine with each other forming H$_2$. In this way, hydrazine coordinated Ni-Co complexes can facilitate the electrocatalysis for HER, thereby enhancing the performance of overall water splitting.

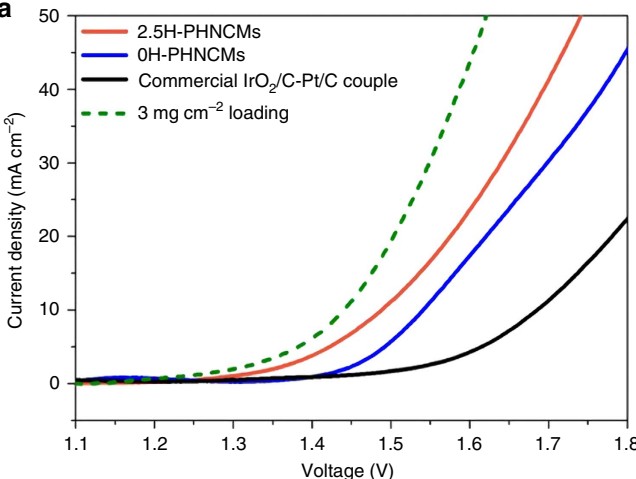

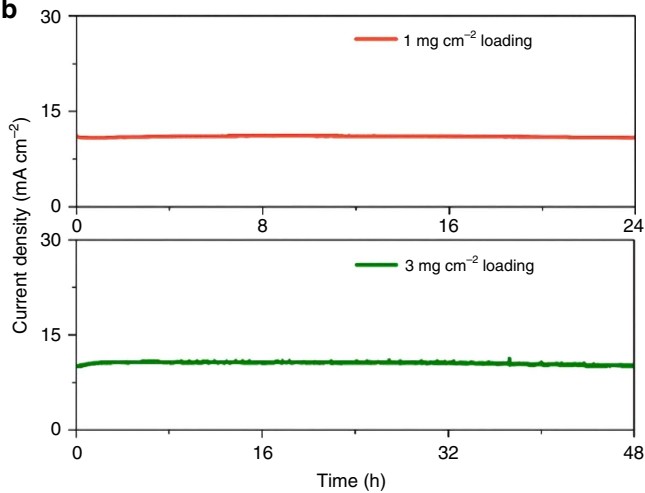

**Figure 5 | Bifunctional electrocatalysts for overall water splitting.** (**a**) Steady-state polarization curves of the catalysts on CFP with 1 mg cm$^{-2}$ of mass loading for overall water splitting at a scan rate of 5 mV s$^{-1}$ in 1 M KOH electrolyte with a two-electrode configuration after a 20 min open circuit scan. The green dash line represents the polarization curve of 2.5H-PHNCMs with 3 mg cm$^{-2}$ of mass loading, which holds just 1.44 V to reach a current density of 10 mA cm$^{-2}$. (**b**) Chronoamperometric curves of 2.5H-PHNCMs with 1 mg cm$^{-2}$ (upper) and 3 mg cm$^{-2}$ (lower) of mass loading for overall water splitting in a two-electrode configuration at constant cell voltages of 1.49 and 1.44 V, respectively.

**Evaluation of PHNCMs for overall water splitting.** PHNCMs were also used as both cathodic and anodic materials for overall water splitting with a two-electrode configuration in 1 M KOH electrolyte. 2.5H-PHNCMs achieved a current density of 10 mA cm$^{-2}$ at a cell voltage of 1.48 V with 1 mg cm$^{-2}$ of mass loading (Fig. 5a). They had higher catalytic activity than 0H-PHNCMs (1.53 V at 10 mA cm$^{-2}$) and commercial IrO$_2$/C-Pt/C couple (1.63 V at 10 mA cm$^{-2}$). According to the polar-ization curves of the PHNCMs (Supplementary Fig. 28a), 2.5H-PHNCMs and 0H-PHNCMs exhibited lower overpotential than that of 0.05H-PHNCMs and 1H-PHNCMs, in accordance with the consequences of HER and OER electrocatalysis. A long-term electrolysis process utilizing 2.5H-PHNCMs was operated at a constant potential of 1.49 V for 24 h (upper image in Fig. 5b), which showed remarkable durability with negligible degradation. However, the current density of commercial IrO$_2$/C-Pt/C couple decreased continuously during a 7 h operation (Supplementary

Fig. 28b). After 24 h operation, many $H_2$ and $O_2$ bubbles remained on the carbon fibre paper (CFP) daubed by 2.5H-PHNCMs with $1\,mg\,cm^{-2}$ of mass loading (Supplementary Fig. 29). In addition, the activity of 2.5H-PHNCMs could be further improved by increasing the mass loading of the active material on CFP to $3\,mg\,cm^{-2}$ (the green dash line in Fig. 5a). The increasing mass loading decreased the overvoltage from 1.48 to 1.44 V at a current density of $10\,mA\,cm^{-2}$. It is noteworthy that this electrode was durable for 48 h without degradation (lower image in Fig. 5b). Furthermore, we investigated the overall water splitting electrocatalytic performances of 3H- and 5H-PHNCMs with $3\,mg\,cm^{-2}$ of mass loading and made a contrast with 2.5H-PHNCMs. The extremely close activities of the three PHNCMs were exhibited in Supplementary Fig. 30. Large volumes of $H_2$ and $O_2$ gases on the surface of electrodes over 2.5H-PHNCMs could be seen during the process of a chronoamperometric test (Supplementary Movie 1, a video of water electrolysis for around 2 min). The above results demonstrated that 2.5H-PHNCMs had a great potential to serve effectively for the practical and long-term application of overall water splitting.

## Discussion

In summary, we have developed a facile approach to fabricate amorphous Ni-Co complexes hybridized with 1T phase $MoS_2$ as highly active bifunctional electrocatalysts for overall water splitting. The hybrid nanostructures exhibit an extremely low overpotential and long-term stability for both HER and OER, which can be attributed to complete conversion of $MoS_2$ to metallic 1T phase, increasing number of catalytic active sites and stabilization effect of amorphous Ni-Co complexes on 1T phase $MoS_2$. This catalyst only needs an overpotential of 1.44 V to afford an overall-water-splitting current density of $10\,mA\,cm^{-2}$ with a mass loading of $3\,mg\,cm^{-2}$ and maintains its catalytic activity for 48 h operation without degradation. The accessible and low-cost manufacturing as well as the excellent electrocatalytic performance may further inspire the development of non-noble-metal electrode materials for overall water splitting.

## Methods

**Reagents.** All the reagents were purchased from Alfa Aesar in analytical grade and used as received without further purification.

**Synthesis of NCUNs.** In a typical synthesis of NCUNs, 0.8 ml of Nickel(II) acetate hydrate aqueous solution (0.2 M) and 0.8 ml of Cobalt(II) acetate tetrahydrate aqueous solution (0.2 M) were added to 8 ml of anhydrous DMF and transferred into a 50 ml-flask. An amount of 74 mg (2 mmol) of ammonium fluoride ($NH_4F$) and 120 mg (2 mmol) of urea ($CO(NH_2)_2$) were dissolved into 0.3 ml of deionized water, respectively. The $NH_4F$ and $CO(NH_2)_2$ aqueous solution were dropwise added into the above mixture with stirring for 24 h. Then the mixture was kept at 90 °C with stirring for 24 h. After the system was cooling down naturally, the suspension was centrifuged and washed with ethanol for three times. Then the precipitant was obtained and re-dispersed in ethanol for further characterizations. Moreover, some precipitant was dried in a freezer dryer with vacuum for the next step reaction. For Ni-only or Co-only hybrids, the precursors were synthesized in the same way above without adding Cobalt(II) acetate tetrahydrate aqueous solution or Nickel(II) acetate hydrate aqueous solution.

**Synthesis of PHNCMs.** First, 25 mg of NCUNs powders and 13 mg of $(NH_4)_2MoS_4$ were dispersed in 10 ml of DMF and sonicated for 10 min to be a homogeneous solution. After sonicating, 0, 0.05 ml, 1 ml, 2.5 ml of HZH was respectively added dropwise into above mixture with vigorously stirring and then 0.1 ml of potassium hydroxide (KOH) aqueous solution (including 20 mg of KOH) was dropwise fed into the mixture with HZH, respectively. Then the mixture was separately transferred into 40 ml Teflon-lined autoclaves and heated at 200 °C for 10 h. After the system was cooling down, the suspension was centrifuged and washed with ethanol for three times. Then the precipitant was obtained and re-dispersed in ethanol for further characterizations. Ni-only or Co-only hybrids were synthesized by the same method above with Ni-only or Co-only precursors.

**Synthesis of $MoS_2$ nanosheets.** A total of 20 mg of $(NH_4)_2MoS_4$ was dispersed in 10 ml of DMF. The mixture was stirred at room temperature for 10 min until a homogeneous solution was achieved. After that, 0, 0.05 ml, 1 and 2.5 ml of HZH were added to four of the above mixture respectively. The reaction solution was further stirred for 10 min before transferred to 40 ml Teflon-lined autoclaves. Then they were heated at 200 °C for 10 h. After the system was cooling down, the suspension was centrifuged and washed with ethanol for three times. Then the precipitant was obtained and re-dispersed in ethanol for further characterizations.

**Characterizations.** The morphologies and structures of the samples were observed by TEM, which was conducted on a Hitachi H7700 at 100 kV, using the carbon-coated copper grid. Details of morphologies and structures were obtained by HRTEM that was carried out on a FEI G2 F20 S-Twin TEM at 200 kV equipped with high angle annular dark-field STEM. EDX were taken with the same instrument as HRTEM. AFM was conducted on a Dimension Icom, Bruker. Powder X-ray diffraction characterization was performed on a Bruker D8 Advance X-ray diffractometer using Cu-Kα radiation ($\lambda = 1.5418$ Å). XPS were recorded on a PHI Quantera SXM spectrometer with monochromatic Al Kα X-ray sources (1,486.6 eV) at 2.0 kV and 20 mA. $N_2$ adsorption/desorption measurements were carried out on a Autosorb-iQ2, Quantachrome Instruments. The proportions of N and Co, Ni, Mo were measured by Elemental Analyzer, Euro EA3000 and ICP-AES, VISTA-MPX, respectively. Ni, Co and Mo K-edge XAFS measurements were made at the beamline 14W1 in 1W1B station in Beijing Synchrotron Radiation Facility (BSRF). The X-ray was monochromatized by a double-crystal Si (111) monochromator for BSRF. The energy was calibrated using a cobalt metal foil for the Co K-edge, a nickel metal foil for the Ni K-edge and a molybdenum metal foil for the Mo K-edge. The monochromator was detuned to reject higher harmonics. The acquired EXAFS data were processed according to the standard procedures using the WinXAS3.1 program[47]. Theoretical amplitudes and phase-shift functions were calculated with the FEFF8.2 code[48] using the crystal structural parameters of the Co, CoO, Ni, NiO, $MoO_3$ and $MoS_2$.

**Preparation of samples for HER and OER electrocatalysis.** 5 mg of the active material and 1 mg of pure carbon (Vulcan XC72) were added into 0.95 ml of mixture solution of water and ethanol with the ratio of 3:1 and sonicated for at least 30 min. When they were dispersed uniformly, 0.05 ml of nafion D-521 dispersion (5% w/w in water and 1-propanol) was added into the above mixture and then sonicated for at least 30 min. After that, 0.006 ml of the mixture solution was drop casted on a rotating disk electrode (RDE) with glassy carbon, which has $0.196\,cm^2$ of effective area. When the sample solution dried naturally, it could be used as the working electrode for HER and OER electrocatalysis.

**Preparation of samples for overall water splitting.** A total of 1 mg of active material, 0.2 mg of acetylene black and 0.15 mg of polyvinylidenefluoride were mixed using 0.04 ml of N-methyl-2-pyrrolidone as the solvent to yield a slurry. Then the slurry was daubed uniformly in $1\,cm^2$ of area on a piece of $1 \times 3\,cm$ CFP and dried in vacuum at 100 °C for 24 h. After that, it was immersed in 1 M KOH solution for 12 h in vacuum for activation. Then it could be used as cathode and anode respectively for overall water splitting and the active area was $1\,cm^2$. The electrode with $3\,mg\,cm^{-2}$ of mass loading was prepared by changing the amount of active material, acetylene black and polyvinylidene difluoride to 3 mg, 0.6 and 0.45 mg, respectively and other procedures were the same as the above.

**Electrochemical characterizations.** HER and OER electrocatalysis were measured in a typical three-electrode configuration with a Princeton PASTAT4000 instrument and a RDE system. Glassy carbon electrode with active sample was selected as the working electrode, SCE as the reference electrode and a graphite rod as the counter electrode. Overall water splitting was performed in a two-electrode system using the CFP with active sample as both the cathode and anode. For HER and OER, all of the polarization curves were measured in 1 M KOH solution using RDE with 1,600 r.p.m. at a scan rate of $5\,mV\,s^{-1}$. Chronoamperometric responses (i-t) were recorded on 2.5H-PHNCMs using RDE with 1,600 r.p.m. for 24 h at a constant applied potential of $-0.13$ V versus RHE for HER and 1.53 V versus RHE for OER. To measure electrochemical double-layer capacitance ($C_{dl}$), the potentials were swept using RDE at 1,600 r.p.m. at a range of no faradic processes six times at six different scan rates (5, 10, 15, 20, 25 and $30\,mV\,s^{-1}$). The measured capacitive current densities at the average potential in the selected range were plotted as a function of the scan rates and the slope of the linear fit could be calculated as the $C_{dl}$. The specific capacitance was generally found to be in the range of $20$–$60\,\mu F\,cm^{-2}$ and we used the average value of $40\,\mu F\,cm^{-2}$ here. According to the following equation[49],

$$ECSA = \frac{C_{dl}}{40\mu F \cdot cm^{-2}}\,cm^2_{ECSA} \qquad (1)$$

we could obtain the ECSA roughly. EIS experiments were performed in the frequency range of 10 kHz–100 mHz at a constant current density of $1\,mA\,cm^{-2}$. All of the Tafel plots were measured in 1 M KOH solution using RDE with

1,600 r.p.m. at a scan rate of $1\,mV\,s^{-1}$. As for the Faradaic efficiency measurements, gas chromatography was used to determine the experimentally evolved amount of $H_2$ and $O_2$. And we used the Faraday law to calculate the theoretical amount of $H_2$ and $O_2$ expected based on a chronopotentiometric response at the constant current density of $20\,mA\,cm^{-2}$ for 120 min. For overall water splitting, all of the polarization curves were measured in 1 M KOH solution at a scan rate of $5\,mV\,s^{-1}$ after an open circuit scan for 20 min. Chronoamperometric curves of 2.5H-PHNCMs with 3 and $1\,mg\,cm^{-2}$ of mass loading and commercial $IrO_2$/C-Pt/C couple were carried out at a constant cell voltage of 1.44, 1.49 and 1.65 V.

**RHE calibration.** In all electrochemical characterizations, SCE was used as the reference electrode and calibrated with RHE. The calibration was taken in the hydrogen-saturated electrolyte with the working electrode of a Pt wire. A single cycle of cyclic voltammetry was measured at a scan rate of $1\,mV\,s^{-1}$, and the average of the two potentials where the current crossed zero was regarded as the potential of the calibration value. In 1 M KOH electrolyte, $E(RHE) = E(SCE) + 1.024\,V$.

**Data availability.** The data reported by this article are available from the corresponding author upon reasonable request.

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

## Acknowledgements

This work was supported by NSFC (21431003, 21521091, 11605201, U1532112), China Ministry of Science and Technology under Contract of 2016YFA0202801 and the State Key Project of Fundamental Research for Nanoscience and Nanotechnology (2014CB848900).

## Author contributions

H. Li. and S.C. contributed equally to this work. X.W. led the research. H. Li. and X.W. conceived the idea. H. Li. planned and performed the experiments, collected the data and analysed the data. S.C. and L.S. collected and analysed the EXAFS data. X.J. proposed and analysed the mechanism of intramolecular proton transfer. H. Li., X.J., B.X., H. Lin, H.Y. and X.W. co-wrote the manuscript. All authors gave approval to the final version of the manuscript.

## Additional information

**Competing interests:** The authors declare no competing financial interests.

