## [Peer Review File · Nature Communications]

Reviewers' Comments:

Reviewer #1 (Remarks to the Author)

The submitted manuscript by Li et al reports the synthesis, characterization, and electrocatalytic water splitting (HER, OER, and overall) performance of Ni-Co complexes hybridized with MoS₂. The manuscript is not recommended for acceptance by Nature Communications because of the following reasons.

There are a large number of transition metal chalcogenides and oxides, including Co, Ni, and Mo-based sulfides and oxides/hydroxides, reported for water splitting applications. The main claim of this hybrid system is the excellent electrocatalytic performance. However, those measured current densities were all calculated with respect to the geometric areas of the assembled electrodes, not directly related to the intrinsic activities of these catalyst samples. It is most likely the higher activity is merely due to the enhanced surface area. In fact, given the high porosity of these catalysts, there must exist substantial capacitance currents. However, the authors failed to account that. In addition, those polarization curves shown in Supplementary Figure 21a look quite odd. Why did 0.05H-PHNCMs and 1H-PHNCMs exhibit such high current densities before 1.5 V (in this case, what is the reference electrode?) while 0H-PHNCMs and 2.5H-PHNCMs showed quite low current densities. Furthermore, why these currents were even negative between 1 and 1.4 V? Did the authors show the cathodic scan or anodic scan?

Many so-called bifunctional electrocatalysts for overall water splitting have been reported these years. Most of those chalcogenides will undergo in situ electrochemical oxidation during OER. However, it is not clear what's going on with those hybridized catalysts during and post OER in the submitted manuscript. Will MoS₂ still remain as metallic 1T-MoS₂?

Another issue this referee finds peculiar is the reactivity order of 0H-PHNCMs, 0.05H-PHNCMs, 1H-PHNCMs, and 2.5H-PHNCMs. What's the real role of hydrazine in impacting the electrocatalytic activities here? Why it decreased the performance first then increase it later? Is 2.5 mL the best situation? How about even more hydrazine was added? Many questions are unclear here. The authors' explanation on the beneficial role of hydrazine in lines 269-283 9 (and Supplementary Figure 20) is rather speculative.

Finally, since hydrazine is a strong reducing agent, why upon addition of hydrazine, both Co and Ni even exhibited more oxidized states rather than reduced states, as implied by the XAS and XPS data? In addition, the quantities of H₂ and O₂ should be measured with gas chromatography. Only seeing bubbles on electrodes is not a convincing evidence of water splitting. The Faradaic efficiencies of both half reactions should be included.

Reviewer #2 (Remarks to the Author)

Overall, this is an excellent piece of work. The authors demonstrated record-type of performance on using the same catalysts for bi-function (HER and OER). The low over potential for both HER and OER is impressive. The stability is pretty good. The authors also carried out extensive characterization to under why the performance is so great. Their explanation is sound. I encourage their publication. The language can later be improved with the editorial help.

Reviewer #3 (Remarks to the Author)

The manuscript reports the synthesis of hybrid nanomaterials consisting of 1T MoS₂ and amorphous NiCo oxide with nanoporosity. In the synthetic design, the addition of hydrazine played an important role in converting crystalline NiCo into amorphous phase, transforming

semiconducting 2H MoS₂ into metallic 1T phase as well as the generation of porous structures within the nanocomposite. Benefited from the numerous active sites of the amorphous NiCo and conductive pathways provided by 1T-MoS₂, the resultant nanocomposite exhibited excellent bifunctionality for oxygen evolution and hydrogen evolution, achieving overall water splitting at low voltage. The formation mechanism and the atomic-level structure of the amorphous NiCo and 1T-phase MoS₂ were explained by a series of control experiments and various characterization techniques (XAFS, XPS, TEM and BET surface analysis). The findings and results presented in this work are significant with original contributions to the advances in the field of electrocatalysis for renewable energy. This work is recommended for publication in Nature Communications after minor revision. Some detailed comments are provided as follows:

1. Hydrazine hydrate was found to be critical in tuning the porosity and the phase of the NiCo and MoS₂ and different amounts of hydrazine were added to study its effects on structure and catalytic performance. The highest amount of hydrazine hydrate studied is 2.5 mL and gives the optimal results. Does higher amount of hydrazine hydrate lead to better results?
2. During the synthesis of the NiCo-MoS₂ nanocomposite, Ni and Co hydroxide ultrathin nanosheets were used as precursors and templates. The crystalline phase of the precursor was identified as Ni(OH)₂ and Co(OH)₂ by XRD as shown in Figure S1. Instead of separate Co and Ni phase, how about NiCo binary oxide as the XRD patterns do not match well the standard? In addition, the peak at $\sim 30^\circ$ and $\sim 45^\circ$ are not assigned.
3. For samples with different amounts of hydrazine hydrate, the HER activity follows the trend 2.5 H-PHNCMs > 0H > 1 H PHNCMs > 0.05 H-PHNCMs. As hydrazine hydrate can increase the porosity, portion of 1T phase MoS₂ and amorphous NiCo phase, the catalytic activity is expected to increase linearly with the amounts of hydrazine hydrate added. What is the reasoning behind the activity trend observed by the authors?
4. The authors believe that the enrichment of amorphous Ni-Co complexes is the main cause for the transformation of 2H MoS₂ to 1T MoS₂. What is the possible reason for this? What roles did the Ni-Co complexes play during such phase conversion?
5. For the ECSA measurements, the surface area values should include both the active material and the conductive carbon additives. The contribution of the conductive carbon should be excluded to give valid ECSA numbers to the active material.

Reply to Referee 1 and revisions made accordingly:

The submitted manuscript by Li et al reports the synthesis, characterization, and electrocatalytic water splitting (HER, OER, and overall) performance of Ni-Co complexes hybridized with MoS₂. The manuscript is not recommended for acceptance by Nature Communications because of the following reasons.

We are grateful to the reviewer's comments. We have made revisions according to each comment, as summarized below.

1. There are a large number of transition metal chalcogenides and oxides, including Co, Ni, and Mo-based sulfides and oxides/hydroxides, reported for water splitting applications. The main claim of this hybrid system is the excellent electrocatalytic performance. However, those measured current densities were all calculated with respect to the geometric areas of the assembled electrodes, not directly related to the intrinsic activities of these catalyst samples. It is most likely the higher activity is merely due to the enhanced surface area. In fact, given the high porosity of these catalysts, there must exist substantial capacitance currents. However, the authors failed to account that.

Reply: Thanks for the comments and suggestions from the reviewer. The enhanced specific surface area can indeed lead to the better electrocatalytic performance. Provided the electrocatalysts exhibited the similar catalytic active sites and centers, the specific surface area should be enlarged as much as possible (it is quite difficult in most cases). But the specific surface area is not the intrinsic factor for electrocatalytic activities. According to the literature (Yu X. Y. *et al. Adv. Mater.* **28**, 9006 (2016).), Yu and co-workers developed a hybrid system of Ni-Co-MoS₂ hollow nanoboxes for enhanced electrocatalytic hydrogen evolution activity. The hollow nanoboxes had 40 m²·g⁻¹ of the specific surface area which was much larger than that of our 0.05H-PHNCMs (26.31 m²·g⁻¹). However, Ni-Co-MoS₂ hollow nanoboxes showed poorer HER electrocatalytic performance than that of our 0.05H-PHNCMs. The overpotentials of Ni-Co-MoS₂ and our 0.05H-PHNCMs at 10 mA/cm² were 155 mV and 148 mV, respectively. Therefore, our PHNCMs had much higher intrinsic electrocatalytic activities.

And then we increased the reaction time and temperature to 48 h and 220 °C during the synthesis of 2.5H-PHNCMs. The resulting morphology exhibited no big differences but some aggregations as compared to the original ones (Fig. R1a). The XRD pattern (Fig. R1b) just matched with the standard MoS₂ peaks (JCPDS No. 37-1492) and displayed the enlargement of interlayer spacing that was in accordance with new (002) and (004) planes with the lattice distance of 0.952 nm and 0.476 nm. This result could be further confirmed in the HRTEM image of the new 2.5H-PHNCMs (Fig. R1c). In the XPS spectrum (Fig. R1d), two characteristic peaks of Mo 3d_{5/2} and Mo 3d_{3/2} orbitals were located at ~228 and ~231.0 eV, which revealed the 1T phase of MoS₂ in the new 2.5H-PHNCMs. However, according to the N₂ adsorption-desorption isotherms (Fig. R1e), the specific surface area of the new 2.5H-PHNCMs was calculated to be 77.86 m²·g⁻¹, which was smaller than that of the original ones. The pore size distribution (Fig. R1f) was also similar with that of the original 2.5H-PHNCMs. Then we evaluated the HER and OER performances of the new 2.5H-PHNCMs and did a comparison with the original ones. As shown in Fig. R2a and R2b, there were no big differences for HER and OER electrocatalytic activities between the new and the original 2.5H-PHNCMs. The gaps between overpotentials at 10 mA/cm² were less than 10 mV for both HER and OER. As for the Tafel slopes

and double-layer capacitances (Fig. R2c-2f), the values were quite close. These results indicated that the specific surface area was not the intrinsic factor in impacting the electrocatalytic activities. Besides, we used a rotating disk electrode with glassy carbon to carry out the HER and OER measurements. The effective geometric area of the rotating disk electrode is 0.196 cm². This value was used to calculate the current densities from the measured currents of all the samples. Therefore, the measured current densities were directly related to the intrinsic activities of these catalyst samples.

As Fig. R12 (seeing the following response of question No.6 of Referee 1) shown, the Faradaic efficiencies of 2.5H-PHNCMs for HER and OER are ~100% and 91.23% (average value), respectively. Such high values indicate that there are slender non-Faradaic currents during the HER and OER measurements, which implies that the high porosity provides a large quantity of electrocatalytic active sites to improve the performance rather than leading to charging currents.

Figure R1. (a) TEM image, (b) XRD pattern, (c) HRTEM image, (d) XPS spectrum of Mo 3d orbitals, (e) N₂ adsorption–desorption isotherms measured at 77 K and (f) Pore size distribution curves of the new 2.5H-PHNCMs synthesized at 220 °C for 48h. Scale bar in (a), 500 nm; (c), 5 nm.

Figure R2. (a) and (b) Polarization curves of the new and original 2.5H-PHNCMs for HER and OER measured at a scan rate of 5 mV/s in 1 M KOH aqueous solution; (c) and (d) the corresponding Tafel plots for HER and OER; (e) and (f) the fitting plots showing C_{dl} of the new 2.5H-PHNCMs for HER and OER. Inset: the corresponding CV scan.

2. In addition, those polarization curves shown in Supplementary Figure 21a look quite odd. Why did 0.05H-PHNCMs and 1H-PHNCMs exhibit such high current densities before 1.5 V (in this case, what is the reference electrode?) while 0H-PHNCMs and 2.5H-PHNCMs showed quite low current densities. Furthermore, why these currents were even negative between 1 and 1.4 V? Did the authors show the cathodic scan or anodic scan?

Reply: We used a two-electrode configuration to measure the overall water splitting performance of our PHNCMs. In this system, two pieces of carbon fiber paper daubed by one sample were used as the cathodic and anodic electrodes respectively. During the measurement, the anodic electrode was measured as the working electrode and the cathodic electrode as both reference electrode and counter electrode. Therefore, the polarization curves for overall water splitting were anodic scan and the currents should be positive in theory. However, at the beginning of the measurement, the samples on the electrodes might be in a non-steady state, especially for Ni-Co-based materials. Therefore, it was very likely for PHNCMs to exhibit the non-zero currents before polarization, which might be positive or negative. Based on this situation, we first stabilized the electrodes using open circuit scan for 20 min and then carried out the polarization measurement at a smaller scan rate (5 mV/s). As Fig. R3 shown, the current densities of the four samples extremely approach to zero before polarization and there were hardly any differences in the electrocatalytic activities of overall water splitting than before. After an open circuit scan, the electrodes preferentially kept a relative balanced state because open circuit voltage was regarded as the steady voltage of electrode. Meanwhile, smaller scan rate was quite beneficial to exhibit the polarization with more stable state.

Figure R3. Steady-state polarization curves comparison for overall water splitting of PHNCMs with 1 mg/cm² of mass loading at a scan rate of 5 mV/s in 1 M KOH electrolyte with a two-electrode configuration after open circuit scan for 20 min.

Revision made: We have changed the Figure 5a and Supplementary Fig. 28a into new ones in the revised manuscript and supplementary information. All of the samples were measured at a scan rate of 5 mV/s after 20-minute open circuit scan for overall water splitting polarization.

3. Many so-called bifunctional electrocatalysts for overall water splitting have been reported these years. Most of those chalcogenides will undergo in situ electrochemical oxidation during OER. However, it is not clear what's going on with those hybridized catalysts during and post OER in the submitted manuscript. Will MoS₂ still remain as metallic 1T-MoS₂?

Reply: As shown in Fig. R4a, two characteristic peaks of Mo 3d_{5/2} and Mo 3d_{3/2} orbitals in 2.5H-PHNCMs after 1000 OER cycles were located at ~228.2 and ~231.0 eV, respectively, which were almost the same as that of 2.5H-PHNCMs before OER. This result confirmed the metallic 1T

phase of MoS₂ in 2.5H-PHNCMs after 1000 OER cycles. Furthermore, new (002) and (004) planes with the enlargement lattice distance of 0.952 nm and 0.476 nm were observed in XRD pattern and HRTEM image of 2.5H-PHNCMs after 1000 OER cycles (Fig. R4c and R4d). The above data indicated that MoS₂ still remained in metallic 1T phase after 1000 OER cycles. When NCUNs were not introduced into the reaction system of 2.5H-PHNCMs, MoS₂ nanosheets with 2.5 ml of hydrazine hydrate (2.5H-MoS₂) were obtained. The positions of the two characteristic peaks of Mo 3d_{5/2} and Mo 3d_{3/2} orbitals in 2.5H-MoS₂ were ~228.7 eV and ~231.9 eV. However, after 1000 OER cycles, upshifts (~0.4 eV) were observed clearly for the peaks of Mo 3d_{5/2} and Mo 3d_{3/2} orbitals in 2.5H-MoS₂, which revealed in situ electrochemical oxidation of 2.5H-MoS₂ during OER process. Therefore, amorphous Ni-Co complexes played a role of stabilization of metallic 1T-MoS₂.

Figure R4. XPS spectra of (a) Mo in 2.5H-PHNCMs and (b) Mo in 2.5H-MoS₂ before and after 1000 OER cycles. (c) XRD pattern and (d) HRTEM image of 2.5H-PHNCMs after 1000 OER cycles. Scale bar in (d), 2 nm.

Revision made: We have added Fig. R4 into the revised supplementary information of the manuscript as Supplementary Fig. 26.

4. Another issue this referee finds peculiar is the reactivity order of 0H-PHNCMs, 0.05H-PHNCMs, 1H-PHNCMs, and 2.5H-PHNCMs. What's the real role of hydrazine in impacting the electrocatalytic activities here? Why it decreased the performance first then increase it later? Is 2.5 mL the best situation? How about even more hydrazine was added? Many questions are unclear here. The authors's explanation on the beneficial role of hydrazine in lines 269-283 9 (and Supplementary Figure 20) is rather speculative.

Reply: Thanks for the comments and suggestions from the reviewer. For the real role of hydrazine, it just could impact the electrocatalytic activities in three aspects indirectly. Firstly, along with the increasing amount of hydrazine hydrate (HZH), MoS₂ in PHNCMs was induced to convert into pure metallic 1T phase from mixed 2H and 1T phase. Metallic 1T-MoS₂ could facilitate the electrode kinetics, increase the electric conductivity of the electrocatalysts and proliferate the catalytic active sites. Secondly, Ni and Co coordinated with HZH forming amorphous complexes. The complexes could stabilize the metallic 1T-MoS₂ and improve the OER and HER performances. Besides, large amount of HZH in the reaction system could create plenty of catalytic active sites and improve the mass transport and gas permeability effectively in the process of water splitting.

As for the reactivity order, it depended on the intrinsic activities and the specific surface area of PHNCMs. For 1H- and 2.5H-PHNCMs, Co and Ni coordinated with HZH forming amorphous complexes and MoS₂ in both of the PHNCMs was in pure 1T phase. Based on the same active centers, 2.5H-PHNCMs had better electrocatalytic activities than 1H-PHNCMs owing to the larger specific surface area and higher porosity. For 0H- and 0.05H-PHNCMs, MoS₂ in both of the two PHNCMs was in the mixed 2H and 1T phase while the amount of Ni and Co metal and the specific surface area of 0H-PHNCMs were quite larger as compared to that of 0.05H-PHNCMs. Therefore, 0H-PHNCMs exhibited superior electrocatalytic performances than 0.05H-PHNCMs because of the better electric conductivity and more active sites. Besides, we also made a comparison of the electrocatalytic activities between 0H- and 1H-PHNCMs. In the beginning of HER, 1H-PHNCMs showed slightly better activity than 0H-PHNCMs. Based on the similar specific surface areas, pure 1T-MoS₂ could be the reason of the easier activation over 1H-PHNCMs for HER. With higher current density, 0H-PHNCMs needed lower potential for operation. According to the FT profiles and the analysis of elemental ratios of Co, Ni and Mo (Fig. 3a, 3b and Supplementary Tab. 3), the ratios of Co and Ni metal in 0H-PHNCMs were higher than that of metallic 1T-MoS₂ in 1H-PHNCMs so that 0H-PHNCMs had better electric conductivity, which led to lower overpotential for operation at the same current over 0H-PHNCMs. This reason could also result in the a little lower overpotential for 0H-PHNCMs in OER electrocatalysis. It could not be denied that specific surface area played a significant role in affecting the electrocatalytic performances. In the reaction system without HZH, it was quite difficult to enlarge the specific surface area. However, the introduction of HZH was an effective way to improve the specific surface area. On the account of similar active center of PHNCMs, larger specific surface area could bring more catalytic active sites to enhance the electrocatalytic performances.

We further increased the amount of HZH to 3 ml and 5 ml (denoted as 3H- and 5H-PHNCMs) during the synthesis of PHNCMs. According to the TEM images (Fig. R5a and R5b), the morphologies and sizes of 3H- and 5H-PHNCMs were almost same as 2.5H-PHNCMs. As combining HRTEM images (Fig. R5c and R5d) and XRD patterns (Fig. R5e), new (002) and (004) planes with the enlarged lattice distance of 0.952 nm and 0.476 nm were observed and pristine (004) planes with the lattice distance of 0.307 nm disappeared. Meanwhile, Ni and Co still kept the amorphous state. The phase of MoS₂ in 3H- and 5H-PHNCMs were still metallic 1T phase, which was further proven in the XPS spectrum of Mo 3d orbitals (Fig. R5f). For both of 3H- and 5H-PHNCMs, the two main peaks of Mo 3d_{3/2} and 3d_{5/2} orbitals located at ~228 eV and 231 eV. We measured N₂ adsorption-desorption at 77 K. Based on the N₂ adsorption-desorption isotherms (Fig. R6a and R6b), the specific surface area were calculated to be 84.98 m²·g⁻¹ and 89.36 m²·g⁻¹, respectively. And the pore distributions of 3H- and 5H-PHNCMs were also semblable to that of 2.5H-PHNCMs (Fig. R6c and R6d), in which pores

with ~ 10 nm diameters were observed. It could be further confirmed in the HAADF-STEM images of 3H- and 5H-PHNCMs (Fig. R6e and R6f). The above data revealed that there were hardly any changes in the morphologies, structures, phase of MoS₂, specific surface area and pore size distributions when increasing amount of hydrazine hydrate. Furthermore, we measured the electrocatalytic activities of 3H- and 5H-PHNCMs and made a comparison with 2.5H-PHNCMs. It was found that the three PHNCMs exhibited quite similar activities for HER and OER electrocatalysis (Fig. R7). Some essential evaluation indexes on HER and OER electrocatalysis were summarized in Tab. R1. The overall water splitting electrocatalysis of 3H- and 5H-PHNCMs with 3 mg/cm² of mass loading using a two-electrode configuration was also carried out (Fig. R8). The extremely close activities of the three PHNCMs were exhibited. In general, all of the three samples showed similar performances in electrocatalysis.

Figure R5. (a) and (b) TEM images, (c) and (d) HRTEM images, (e) XRD patterns, (f) XPS spectrum of 3H- and 5H-PHNCMs. Scale bars in (a) and (b), 200 nm; (c) and (d), 5 nm.

Figure R6. N₂ adsorption–desorption isotherms measured at 77 K of (a) 3H- and (b) 5H-PHNCMs. The specific surface areas were calculated to be 84.98 m²·g⁻¹ and 89.36 m²·g⁻¹, respectively. (c) and (d) The corresponding pore size distribution curves of 3H- and 5H-PHNCMs. (e) and (f) HADDF-STEM images of 3H- and 5H-PHNCMs. Scale bars in (e) and (f), 20 nm.

Electrocatalytic reaction	Samples	Overpotential at 10 mA·cm ⁻² (V vs RHE)	Tafel slope (mV·decade ⁻¹)	The double-layer capacitance (mF·cm ⁻²)
HER	2.5H-PHNCMs	0.07	38.1	32.3
	3H-PHNCMs	0.075	38.7	31.8
	5H-PHNCMs	0.08	40.3	32.3
OER	2.5H-PHNCMs	1.465	45.7	108.1
	3H-PHNCMs	1.472	46.0	109.8
	5H-PHNCMs	1.474	46.2	109.4

Table R1. Evaluation indexes of 2.5H-, 3H- and 5H-PHNCMs for HER and OER electrocatalytic activities.

Figure R7. (a) and (b) Polarization curves of 2.5H-, 3H- and 5H-PHNCMs for HER and OER measured at a scan rate of 5 mV/s in 1 M KOH aqueous solution; (c) and (d) the corresponding Tafel plots for HER and OER; (e) and (f) the fitting plots showing C_{dl} for HER and OER. Inset: the corresponding CV scan.

Figure R8. Steady-state polarization curves of the samples on carbon fiber paper with 3 mg/cm² of mass loading for overall water splitting in 1 M KOH electrolyte at a scan rate of 5mV/s with a two-electrode configuration after an open circuit scan for 20 min.

The mechanism of intramolecular proton transfer in hydrazine involved Ni and Co complexes was just put forward based on the previous reports (Wilson, A. D. et al. *J. Am. Chem. Soc.* 128, 358-366 (2006); Jacques P. A. et al. *Proc. Natl. Acad. Sci.* 106, 20627–20632 (2009).) and reasonable speculation. Our amorphous Ni-Co complexes have amine residues in the second coordination sphere, which could be a hydrogen-exchanging site. According to the fitting results of XPS spectra of Ni and Co 2p orbitals in 2.5H-PHNCMs (Fig. R9, seeing the response of next question raised by Referee 1), Co (III) and Ni (III) predominated in 2.5H-PHNCMs. These trivalent Co and Ni linking to hydrazine lose two electrons to become monovalent metals firstly. Then, two protons combine with the monovalent metal and N of the amine residue in the second coordination sphere respectively to form a metal hydride and a $-\text{NH}_3^+$ group. Subsequently, the two protons tended to combine with each other forming H_2 . It was just an extremely possible way for hydrazine involved Ni-Co complexes to enhance the performance of HER.

Revision made: We have added new Supplementary figures into the revised Supplementary information of the manuscript as Supplementary Fig. 5, 6e, 6f, 7, 16 to demonstrate the details of morphologies, specific surface areas, pore size distributions, crystalline structures and the phase of MoS_2 in 3H- and 5H-PHNCMs. Supplementary Fig. 25, Supplementary Tab. 5 and Supplementary Fig. 30 were added to exhibited the electrocatalytic activities of 3H- and 5H-PHNCMs, including HER, OER and overall water splitting.

5. Finally, since hydrazine is a strong reducing agent, why upon addition of hydrazine, both Co and Ni even exhibited more oxidized states rather than reduced states, as implied by the XAS and XPS data?

Reply: As Fig. R9 shown, Co (III) and Ni (III) predominated along with Co (II) and Ni (II) coexisting on the surface of 2.5H-PHNCMs. But there were just a small quantity of Co and Ni metal on the surface of 2.5H-PHNCMs due to the small peaks observed in the XPS spectra. According to the XANES data (Fig. 3a and 3b), Ni and Co in the oxidation states were the major component in the 2.5H-PHNCMs. During the synthesis of 2.5H-PHNCMs, hydrazine was added dropwise slowly into the reaction solution with stirring in the room temperature before heating. In this case, Co and Ni ions preferentially coordinated with hydrazine to form complexes first rather than being reduced owing to the strong coordination ability of hydrazine. Meanwhile, a small quantity of potassium hydroxide (KOH) was added into the system to make NH_3 form and escape easily at a high temperature. The NH_3 would be a gas template to make sure the formation of porous structures. The amount of KOH was not enough to make the preformed Ni-Co-hydrazine complexes to be reduced to metallic states (Nicholls, D. et al. *J. Chem. Soc. A* 950-952 (1966); Park, J. W. et al. *Mater. Chem. Phys.* **97**, 371-378 (2006)). Therefore, the amount of Co and Ni metal was inappreciable in 2.5H-PHNCMs. When Co and Ni ions coordinated with hydrazine to form complexes, it is easy for Co (II) and Ni (II) complexes to be oxidized to Co (III) and Ni (III) complexes because the reduction potential of hydrazine-coordinated complexes, $\psi^0(\text{Co}^{3+}/\text{Co}^{2+})$ and $\psi^0(\text{Ni}^{3+}/\text{Ni}^{2+})$, was decreased dramatically to the 10^{-1} of magnitude. Therefore, Co (III) and Ni (III) hydrazine-coordinated complexes could exist steadily. In conclusion, both Co and Ni even exhibited more oxidized states rather than reduced states.

Figure R9. The fitting results of XPS spectra of (a) Co 2p and (b) Ni 2p orbitals in 2.5H-PHNCMs.

Revision made: We have added the Figure R9 into the revised supplementary information of the manuscript as Supplementary Fig. 15.

6. *In addition, the quantities of H_2 and O_2 should be measured with gas chromatography. Only seeing bubbles on electrodes is not a convincing evidence of water splitting. The Faradaic efficiencies of both half reactions should be included.*

Reply: Thanks for the reviewer's kind suggestion. We used the gas chromatography to measure the amount of H_2 and O_2 during 2-hour processes of chronoamperometric responses at the constant current density of 20 mA/cm^2 . As shown in Fig. R10a, 2.5H-PHNCMs exhibited a gradually increasing hydrogen evolution process that extremely approached to the theoretical values. This result indicated that 2.5H-PHNCMs provided a Faradaic efficiency of $\sim 100\%$ for HER. It is a convincing evidence of water splitting, which means that the cathodic current over 2.5H-PHNCMs derives from the hydrogen evolution.

The measured values of O₂ production were slightly less than the theoretical values (Fig. R10b) due to the complex four-electron reaction process, whose kinetics was quite unfavourable. The comparison of the theoretical values and measured values of O₂ evolution at different periods was summarized in Table R2 and the Faradaic efficiencies were also calculated. The average Faradaic efficiency was 91.23%.

Figure R10. (a) and (b) Electrocatalytic Faradaic efficiencies of HER and OER using 2.5H-PHNCMs as the active material on a piece of carbon fiber paper (effective electrode area: 0.5 cm²) at a current density of 20 mA/cm² and measured for 120 min.

Time (min)	10	20	30	40	50	60	70	80	90	100	110	120
Theoretical value (µmol)	15.55	31.09	46.64	62.19	77.73	93.28	108.83	124.37	139.92	155.46	171.01	186.56
Measured value (µmol)	14.12	28.36	42.58	56.90	71.12	85.26	99.58	113.55	127.61	141.47	155.79	169.58
Faradaic efficiency (%)	90.8	91.2	91.3	91.5	91.5	91.4	91.5	91.3	91.2	91	91.1	90.9

Table R2. The theoretical values, measured values and the Faradaic efficiencies of the amount of O₂ production at different periods.

Revision made: We have added the Fig. R10 and Table R2 into the revised supplementary information of the manuscript as the Supplementary Fig. 23 and Supplementary Table 4.

We thank the reviewer for the insightful comments and kind suggestions! The reply for each question/comment is expected to reach the high criteria.

Reply to Referee 2 and revisions made accordingly:

Overall, this is an excellent piece of work. The authors demonstrated record-type of performance on using the same catalysts for bi-function (HER and OER). The low over potential for both HER and OER is impressive. The stability is pretty good. The authors also carried out extensive characterization to under why the performance is so great. Their explanation is sound. I encourage their publication. The language can later be improved with the editorial help.

Reply: Thanks for the reviewer's positive evaluation of this work and suggestion. We have corrected the syntax errors and improved the language expression in the revised manuscript.

Reply to Referee 3 and revisions made accordingly:

The manuscript reports the synthesis of hybrid nanomaterials consisting of 1T MoS₂ and amorphous NiCo oxide with nanoporosity. In the synthetic design, the addition of hydrazine played an important role in converting crystalline NiCo into amorphous phase, transforming semiconducting 2H MoS₂ into metallic 1T phase as well as the generation of porous structures within the nanocomposite. Benefited from the numerous active sites of the amorphous NiCo and conductive pathways provided by 1T-MoS₂, the resultant nanocomposite exhibited excellent bifunctionality for oxygen evolution and hydrogen evolution, achieving overall water splitting at low voltage. The formation mechanism and the atomic-level structure of the amorphous NiCo and 1T-phase MoS₂ were explained by a series of control experiments and various characterization techniques (XAFS, XPS, TEM and BET surface analysis). The findings and results presented in this work are significant with original contributions to the advances in the field of electrocatalysis for renewable energy. This work is recommended for publication in Nature Communications after minor revision. Some detailed comments are provided as follows:

We are grateful to the reviewer's comments. We have made revisions according to each comment, as summarized below.

1. Hydrazine hydrate was found to be critical in tuning the porosity and the phase of the NiCo and MoS₂ and different amounts of hydrazine were added to study its effects on structure and catalytic performance. The highest amount of hydrazine hydrate studied is 2.5 mL and gives the optimal results. Does higher amount of hydrazine hydrate lead to better results?

Reply: Thanks for the comments and suggestions from the reviewer. We further increased the amount of HZH to 3 ml and 5 ml (denoted as 3H- and 5H-PHNCMs) to investigate the structures and electrocatalytic performances. According to the TEM images (Fig. R11a and R11b), the morphologies and sizes of 3H- and 5H-PHNCMs were almost same as 2.5H-PHNCMs. As combining HRTEM images (Fig. R11c and R11d) and XRD patterns (Fig. R11e), new (002) and (004) planes with enlarged lattice distance of 0.952 nm and 0.476 nm were observed and pristine (004) planes with the lattice distance of 0.307 nm disappeared. Meanwhile, Ni and Co still kept the amorphous state. The phase of MoS₂ in 3H- and 5H-PHNCMs were still metallic 1T phase, which was proven in the XPS spectrum of Mo 3d orbitals (Fig. R11f). For both of 3H- and 5H-PHNCMs, the two main peaks of Mo 3d_{3/2} and 3d_{5/2} orbitals located at ~228 eV and 231 eV. For the specific surface area and pore size distribution, we measured N₂ adsorption-desorption at 77 K. Based on the N₂ adsorption-desorption isotherms (Fig. R12a and R12b), the specific surface area were calculated to be 84.98 m²·g⁻¹ and 89.36 m²·g⁻¹, respectively. And the pore distributions of 3H- and 5H-PHNCMs were also similar to that of 2.5H-PHNCMs (Fig. R12c and R12d), in which pores with ~10 nm diameters were observed, which could be further confirmed in the HAADF-STEM images of 3H- and

5H-PHNCMs (Fig. R12e and R12f). The above data revealed that there were hardly any changes in the morphologies, structures, phase of MoS₂, specific surface area and pore size distributions when increasing amount of hydrazine hydrate.

We also measured the electrocatalytic activities of 3H- and 5H-PHNCMs and made a comparison with 2.5H-PHNCMs. It was found that the three PHNCMs exhibited quite similar activities for HER and OER electrocatalysis (Fig. R13). Besides, some evaluation indexes on HER and OER electrocatalysis were summarized in Tab. R3. Furthermore, we investigated the overall water splitting electrocatalysis of 3H- and 5H-PHNCMs with 3 mg/cm² of mass loading using a two-electrode configuration (Fig. R14). Their activities of overall water splitting exhibited an extremely close result as compared to 2.5H-PHNCMs. In general, all of the three samples showed similar performances in electrocatalysis.

Figure R11. (a) and (b) TEM images, (c) and (d) HRTEM images, (e) XRD patterns, (f) XPS spectrum of 3H- and 5H-PHNCMs. Scale bars in (a) and (b), 200 nm; (c) and (d), 5 nm.

Figure R12. N_2 adsorption–desorption isotherms measured at 77 K of (a) 3H- and (b) 5H-PHNCMs. The specific surface areas were calculated to be $84.98 \text{ m}^2 \cdot \text{g}^{-1}$ and $89.36 \text{ m}^2 \cdot \text{g}^{-1}$, respectively. (c) and (d) The corresponding pore size distribution curves of 3H- and 5H-PHNCMs. (e) and (f) HADDF-STEM images of 3H- and 5H-PHNCMs. Scale bars in (e) and (f), 20 nm.

Electrocatalytic reaction	Samples	Overpotential at $10 \text{ mA} \cdot \text{cm}^{-2}$ (V vs RHE)	Tafel slope ($\text{mV} \cdot \text{decade}^{-1}$)	The double-layer capacitance ($\text{mF} \cdot \text{cm}^{-2}$)
HER	2.5H-PHNCMs	0.07	38.1	32.3
	3H-PHNCMs	0.075	38.7	31.8
	5H-PHNCMs	0.08	40.3	32.3
OER	2.5H-PHNCMs	1.465	45.7	108.1
	3H-PHNCMs	1.472	46.0	109.8
	5H-PHNCMs	1.474	46.2	109.4

Table R3. Evaluation indexes of 2.5H-, 3H- and 5H-PHNCMs for HER and OER electrocatalytic activities.

Figure R13. (a) and (b) Polarization curves of 2.5H-, 3H- and 5H-PHNCMs for HER and OER measured at a scan rate of 5 mV/s in 1 M KOH aqueous solution; (c) and (d) the corresponding Tafel plots for HER and OER; (e) and (f) the fitting plots showing C_{dl} for HER and OER. Inset: the corresponding CV scan.

Figure R14. Steady-state polarization curves of the samples on carbon fiber paper with 3 mg/cm² of mass loading for overall water splitting in 1 M KOH electrolyte at a scan rate of 5mV/s with a two-electrode configuration after an open circuit scan for 20 min.

Revision made: We added new Supplementary figures into the revised Supplementary information of the manuscript as Supplementary Fig. 5, 6e, 6f, 7, 16 to demonstrate the details of morphologies, specific surface areas, pore size distributions, crystalline structures and the phase of MoS₂ in 3H- and 5H-PHNCMs. Supplementary Fig. 25, Supplementary Tab. 5 and Supplementary Fig. 30 were added to exhibited the electrocatalytic activities of 3H- and 5H-PHNCMs, including HER, OER and overall water splitting.

2. During the synthesis of the NiCo-MoS₂ nanocomposite, Ni and Co hydroxide ultrathin nanosheets were used as precursors and templates. The crystalline phase of the precursor was identified as Ni(OH)₂ and Co(OH)₂ by XRD as shown in Figure S1. Instead of separate Co and Ni phase, how about NiCo binary oxide as the XRD patterns do not match well the standard? In addition, the peak at ~ 30° and ~ 45° are not assigned.

Reply: Thanks for the reviewer's kind suggestion. Instead of separate Co and Ni phase, we tried to use the standard NiCo₂O₄ peaks (JCPDS No. 20-0781) to match with the XRD pattern of NCUNs (Fig. R15a). Even though there were two peaks at ~ 30° and ~ 45° of standard NiCo₂O₄ peaks, the XRD pattern of NCUNs still did not match well with them. For the standard Ni(OH)₂ peaks (JCPDS No.38-0715), there was a peak at 45.98° that could match with that of the XRD pattern. Furthermore, the XPS spectrum (Fig. R15b) of O 1s orbital in NCUNs was exhibited to prove that NCUNs were composed of hydroxides. The peak was located at 531.5 eV, which was the position of characteristic peak of O 1s orbital in hydroxides (Tan B. J., Klabunde K. J., Sherwood P. M. A., *Chem. Mater.* **2**, 186–191 (1990)).

Figure R15. (a) XRD pattern of NCUNs. The violet lines present the standard nickel hydroxide (JCPDS No. 38-0715) peaks, the olive lines demonstrate the standard cobalt hydroxide (JCPDS No. 20-0781) peaks. (b) XPS spectrum of O 1s orbital in NCUNs.

Revision made: We have changed the Supplementary Fig. 1a into a new one with adding the standard Ni(OH)₂ peaks (JCPDS No. 38-0715). The XPS spectrum of O 1s orbital in NCUNs has been also added into the revised supplementary information of the manuscript as Supplementary Fig. 1b. Besides, the literature (Tan B. J., Klabunde K. J., Sherwood P. M. A., *Chem. Mater.* **2**, 186–191 (1990).) has been added into the References.

3. For samples with different amounts of hydrazine hydrate, the HER activity follows the trend 2.5 H-PHNCMs > 0H > 1 H PHNCMs > 0.05 H-PHNCMs. As hydrazine hydrate can increase the porosity, portion of 1T phase MoS₂ and amorphous NiCo phase, the catalytic activity is expected to increase linearly with the amounts of hydrazine hydrate added. What is the reasoning behind the activity trend observed by the authors?

Reply: For the activity trend, it depended on the intrinsic activities and the specific surface area of PHNCMs. For 1H- and 2.5H-PHNCMs, Co and Ni coordinated with HZH forming amorphous complexes and MoS₂ in both of the PHNCMs was in pure 1T phase. Based on the same active centers, 2.5H-PHNCMs had better electrocatalytic activities than 1H-PHNCMs owing to the larger specific surface area and higher porosity. For 0H- and 0.05H-PHNCMs, MoS₂ in both of the two PHNCMs was in the mixed 2H and 1T phase while the amount of Ni and Co metal and the specific surface area of 0H-PHNCMs were quite larger as compared to that of 0.05H-PHNCMs. Therefore, 0H-PHNCMs exhibited superior electrocatalytic performances than 0.05H-PHNCMs because of the better electric conductivity and more active sites. Besides, we also made a comparison of the electrocatalytic activities between 0H- and 1H-PHNCMs. In the beginning of HER, 1H-PHNCMs showed slightly better activity than 0H-PHNCMs. Based on the similar specific surface areas, pure 1T-MoS₂ could be the reason of the easier activation over 1H-PHNCMs for HER. With higher current density, 0H-PHNCMs needed lower potential for operation. According to the FT profiles and the analysis of elemental ratios of Co, Ni and Mo (Fig. 3a, 3b and Supplementary Tab. 3), the ratios of Co and Ni metal in 0H-PHNCMs were higher than that of metallic 1T-MoS₂ in 1H-PHNCMs so that 0H-PHNCMs had better electric conductivity, which led to lower overpotential for operation at the same current over 0H-PHNCMs. This reason could also result in the a little lower overpotential for 0H-PHNCMs in OER electrocatalysis. It could not be denied that specific surface area played a significant role in affecting the electrocatalytic performances. In the reaction system without HZH, it was quite difficult to enlarge the specific surface area. However, the introduction of HZH was an effective way to improve the specific surface area. On the account of similar active center of PHNCMs, larger specific surface area could bring more catalytic active sites to enhance the electrocatalytic performances.

4. The authors believe that the enrichment of amorphous Ni-Co complexes is the main cause for the transformation of 2H MoS₂ to 1T MoS₂. What is the possible reason for this? What roles did the Ni-Co complexes play during such phase conversion?

Reply: Thanks for the comments and suggestions from the reviewer. It is well known that charge transfer from intercalated lithium to the MoS₂ host lattice can induce a first order phase transition in which the Mo coordination changes from a direct band gap semiconducting trigonal prismatic (2H) to a metallic octahedral (1T) structure (Conley H. J. *et al. Nano Lett.* **13**, 3626-3630 (2013)). Therefore, electron donor is quite necessary in the phase transformation of MoS₂ from 2H to 1T phase. In our case, hydrazine could be regarded as the electron donor to induce the phase transformation. However,

if NCUNs were not introduced to the reaction system of 2.5H-PHNCMs, pure 1T-MoS₂ could not be obtained according to the XPS spectrum of Mo in MoS₂ synthesized in the same way as 2.5H-PHNCMs (Fig. 3f). Compared to 2H phase MoS₂, 1T-MoS₂ has higher ground-state energy according to the literature (Reed E. J. *et al. Nat. Commun.* **5**, 4214-4222 (2014).). Therefore, 1T-MoS₂ needs to be stabilized. It is extremely possible that amorphous Ni-Co complexes play a role of stabilization of 1T-MoS₂. Metal chalcogenides preferentially undergo in situ electrochemical oxidation during OER. We made a comparison of XPS spectra (Fig. R15a and R15b) of Mo in 2.5H-PHNCMs and 2.5H-MoS₂ before and after 1000 OER cycles. It was found that the positions of the two characteristic peaks of Mo 3d_{5/2} and Mo 3d_{3/2} orbitals in 2.5H-PHNCMs did not change after 1000 OER cycles. However, upshifts (~0.4 eV) were observed for the peaks of Mo 3d_{5/2} and Mo 3d_{3/2} orbitals in 2.5H-MoS₂ after 1000 OER cycles, which revealed in situ electrochemical oxidation of 2.5H-MoS₂ during OER process. Besides, the enlargement of new (002) and (004) planes with the lattice distance of 0.952 nm and 0.476 nm were observed in the XRD pattern (Fig. R15c) and HRTEM image (Fig. R15d) of 2.5H-PHNCMs after 1000 OER cycles, which could be further confirmed that MoS₂ in 2.5H-PHNCMs remained 1T phase after 1000 OER cycles. Therefore, amorphous Ni-Co complexes could stabilize 1T-MoS₂ and the enrichment of amorphous Ni-Co complexes is the main cause for the transformation of 2H- to 1T-MoS₂.

Figure R15. XPS spectra of (a) Mo in 2.5H-PHNCMs and (b) Mo in 2.5H-MoS₂ before and after 1000 OER cycles. (c) XRD pattern and (d) HRTEM image of 2.5H-PHNCMs after 1000 OER cycles. Scale bar in (d), 2 nm.

Revision made: We have added Fig. R15 into the revised supplementary information of the manuscript as Supplementary Fig. 26 and two new literatures (Conley H. J. *et al. Nano Lett.* **13**,

3626-3630 (2013); Reed E. J. *et al. Nat. Commun.* **5**, 4214-4222 (2014).) were added into the References in the revised manuscript as the reference 43 and 44.

5. For the ECSA measurements, the surface area values should include both the active material and the conductive carbon additives. The contribution of the conductive carbon should be excluded to give valid ECSA numbers to the active material.

Reply: Thanks for the reviewer's kind suggestion. It is well known that pure carbon preferentially increases the conductivity and dispersity of the samples so that higher current densities and lower overpotentials for electrocatalysis will be exhibited. But the pure carbon we used, vulcan XC72, can hardly be polarized for both HER and OER (seeing the black curves in Fig. R16a and R16b). Therefore, the intrinsic performances of the pure carbon for both HER and OER can be neglected. As the polarization curves of HER and OER (Fig. R16a and Fig. R16b) shown, the samples without adding pure carbon exhibited the weaker performances than that with pure carbon (Fig. 4a and 4b), owing to the deteriorative conductivity and dispersity. We measured the double-layer capacitance (C_{dl}) of the four samples without the pure carbon (Fig. R16c and R16d). The potentials were swept for a cycle using RDE with 1600 rpm in a range of no faradic processes for six times at six different scan rates. The current densities at the average potential in the selected range were plotted as a function of the scan rates and the slope of the linear fit could be calculated as the C_{dl} . The ECSA values of the samples for both HER and OER were shown in Table R4. We found that the ECSA became smaller as compared to that of the samples with conductive carbon, which indicated that the conductive carbon could increase the catalytic active sites without impacting the intrinsic performances of the samples.

Figure R19. (a) and (b) Polarization curves of the samples without adding conductive carbon for HER and OER measured at a scan rate of $5 \text{ mV}\cdot\text{s}^{-1}$ in 1 M KOH; (c) and (d) the fitting plots showing C_{dl} of the samples without adding conductive carbon for HER and OER.

		2.5H-PHNCMs	0H-PHNCMs	MoS₂ nanosheets	NCUNs
HER	ECSA	545	260	152.5	17.5
(cm²)					
OER	ECSA	1930	1202.5	570	22.5
(cm²)					

Table R4. The ECSA values of the samples without adding conductive carbon for HER and OER.

We thank the reviewer for the insightful comments and kind suggestions! The reply for each question/comment is expected to reach the high criteria.

Reviewers' Comments:

Reviewer #1 (Remarks to the Author)

The authors have addressed most of the concerns raised by the previous referees. This revised version is acceptable to Nature Communications now.

Reviewer #3 (Remarks to the Author)

The revision has fully addressed my questions. I enthusiastically support the publication of the manuscript in Nature Communications with no further delay.